# DATA FORGING ATTACKS ON CRYPTOGRAPHIC MODEL CERTIFICATION

## ABSTRACT

Privacy-preserving machine learning auditing protocols allow auditors to assess models for properties such as fairness or robustness, without revealing their internals or training data. This makes them especially attractive for auditing models deployed in sensitive domains such as healthcare or finance. For these protocols to be truly useful, though, their guarantees must reflect how the model will behave once deployed, not just under the conditions of an audit. Existing security definitions often miss this mark: most certify model behavior only on a *fixed audit dataset*, without ensuring that the same guarantees *generalize* to other datasets drawn from the same distribution. We show that a model provider can attack many cryptographic model certification schemes by forging training data, resulting in a model that exhibits benign behavior during an audit, but pathological behavior in practice. For example, we empirically demonstrate that an attacker can train a model that achieves over 99% accuracy on an audit dataset, but less than 30% accuracy on fresh samples from the same distribution.

To address this gap, we formalize the guarantees an auditing framework should achieve and introduce a generic protocol template that meets these requirements. Our results thus offer both cautionary evidence about existing approaches and constructive guidance for designing secure, privacy-preserving ML auditing protocols.

## 1 INTRODUCTION

Certifiable, privacy-preserving machine learning aims to formally prove desired properties of the model while keeping model parameters and training data confidential (Zhang et al., 2020; Liu et al., 2021; Shamsabadi et al., 2022). In this context, the typical lifecycle follows a sequence in which the model provider first trains the model, then an auditor evaluates it according to desired criteria, and—after passing the audit —the certified model is deployed.[1] Note that certification comes from the use of cryptography (e.g., cryptographic commitments, zero-knowledge proofs (Goldwasser et al., 1985)) rather than a specific ML algorithm. The usage of cryptographic techniques allows to not only certify the intended property, but do so while keeping the model internals and training data private. However, it turns out that the guarantees that model certifications provide are bound to the specific dataset that was used during the audit (e.g., "a demographic parity gap of the model held by the provider is below 10% on the UCI Default Credit dataset"). In this paper, we observe that such dataset-specific guarantees risk creating a false sense of security: by themselves, they do not ensure that the certified properties will continue to hold once the model is deployed and applied to *fresh data*, even when this data is drawn from the same distribution as the audit dataset. We show that this is not merely a theoretical concern.

We propose novel attack strategies allowing an adversarial model provider to pass an audit (thus enabling deployment) while simultaneously pursuing its own, potentially conflicting, interests. For example, while an auditor may seek to verify fairness, the model owner may instead prioritize accuracy—even when accuracy and fairness are in tension. We show that when the audit dataset is known in advance (as is often the case when public benchmark datasets are used), the model owner can carefully engineer "training data" so that a model honestly trained on it passes the audit, while exhibiting pathological behavior on real-world inputs. We empirically show that such **data forging**

---

[1]Some works require continuous auditing during deployment instead of a single audit pre-deployment; see Table 1.

Table 1: Analysis of vulnerabilities to data-forging attacks in privacy-preserving ML audits. ✓= supported; ▲= conditional; ✗= not supported.

| Work | Certified property | | | | Resilience to data-forging | Continuous verification |
|---|---|---|---|---|---|---|
| | Acc. | Group Fair | Indv. Fair | Diff. Priv. | | |
| Zhang et al. (2020) | ✓ | ✗ | ✗ | ✗ | ▲ (pd) | ✗ |
| Shamsabadi et al. (2022) | ✗ | ✓ | ✗ | ✗ | ✗ | ✗ |
| Yadav et al. (2024) | ✗ | ✗ | ✓ | ✗ | ✓ | ✓ |
| Liu et al. (2021) | ✓ | ✗ | ✗ | ✗ | ▲ (pd) | ✗ |
| Franzese et al. (2024) | ✗ | ✓ | ✗ | ✗ | ✓ | ✓ |
| Shamsabadi et al. (2024) | ✗ | ✗ | ✗ | ✓ | ✗ | ✗ |
| Kang et al. (2022) | ✓ | ✗ | ✗ | ✗ | ✓ | ✗ |
| Wang and Hoang (2023) | ✓ | ✗ | ✗ | ✗ | ▲ (pd) | ✗ |
| Bourrée et al. (2025) | ✗ | ✓ | ✗ | ✗ | ✗ | ✗ |

Acc. = accuracy; Group/Indv. Fair = group/individual fairness; Diff. Priv.=differential privacy. "Conditional" works lack detail to assess resilience to data-forging, but indicate deployments with public datasets (pd), which would be make the solution vulnerable. Continuous verification means audits must run continuously during deployment (e.g., via clients) rather than once pre-deployment.

**attacks** can cause dramatic gaps between audit-time guarantees and true model performance: for instance, in one of our attacks a model can pass an audit requiring 80% accuracy on the audit dataset, yet achieve only 30% accuracy on new samples from the same distribution. We establish the attacks rigorously for decision trees—both empirically and formally—and provide preliminary empirical evidence for neural networks. We show that our attacks remain undetected by straightforward approaches such as statistical tests, e.g., Welch's $t$-test (Welch, 1947) are performed to check whether the training data and audit data were taken from the same distribution. We further show that a number of prior works are vulnerable to such data forging attacks (see Table 1).

Motivated by these vulnerabilities, we introduce a formal foundation for certifiable machine learning. This includes a formal security definition ensuring that a model provider passes the audit if and only if the model has the desirable property on a given data distribution. We further formalize an attack game that highlights the gap between certifying a property on a fixed dataset and certifying that the same property generalizes to fresh samples from the distribution. Finally, we propose a generic method for achieving secure machine learning auditing. Our approach is agnostic to the specific property that is being certified and, as we formally prove, guarantees that whenever a model passes an audit, the certified properties will also hold at deployment. The key ingredient is ensuring that the audit is conducted on test data that is *independent* of both the model and its training data. This method might serve as a template for future works to obtain not only efficient, but also secure auditing solutions.

In summary, our work advances the study of cryptographic auditing for machine learning by **(i)** proposing a novel attack strategy that passes an audit while enabling pathological model behavior at deployment with respect to real-world inputs; **(ii)** empirically demonstrating the effectiveness of our attack against three example certification objectives: accuracy auditing, fairness auditing, and statistics for distribution similarity testing; **(iii)** introducing formal security definitions tailored to certifiable machine learning and a protocol template that mitigates the attack. We emphasize that we *do not suggest that prior cryptographic works are broken on a technical level*, rather that the guarantees these works provide deserve closer scrutiny. Our findings comprise strong evidence that secure audit solutions with any of the following properties are unlikely: a) those which utilize known public datasets for test purposes, and b) those that reuse test datasets (if model owner learns a substantial amount of this test dataset during the audit). This evidences the importance of continuous sampling of fresh data for a successful audit infrastructure. We hope that our work will inform the design of future cryptographically secure machine learning audit frameworks.

## 2 RELATED WORK

Our work is related to, but distinct from, data poisoning attacks (Steinhardt et al., 2017). Such attacks have traditionally been considered in the context of machine learning systems trained on user-provided

data. Both data poisoning attacks and the concrete attacks in our work (see §4.1) involve adversarial manipulations of training data. However, the data poisoning setting is conceptually different from ours: In data poisoning, the model provider is typically considered honest, and the concern is that users contributing to the model can inject malicious data to degrade a model's performance. As a result, data poisoning involves subtle, often small-scale perturbations to the training data. More formally, data poisoning can be viewed as a game between a *defender*, who seeks to learn an accurate model, and an *attacker*, who wishes to corrupt the learned model (Barreno et al., 2010). The model is honestly trained on the combination of a clean dataset $D_c$ and a poisoned dataset $D_p$, where the size of $D_p$ is no larger than that of $D_c$. In contrast, we consider a fully malicious model provider. Its goal is to engineer a model that passes an audit, while violating the certified properties on real-world data. Our adversary is not restricted to small-scale perturbations of the clean training data and is not required to perform the training in an honest way.

The conclusions we draw about requiring fresh data for auditing are semantically related to work on the inadequacy of public benchmarks in machine learning Zhang et al. (2025a); Hardt (2025), but those works do not consider cryptographic security. For additional related work and an overview of certifiable ML, see §F.

## 3   CERTIFYING ML: BACKGROUND AND UNIFYING SYNTAX

Consider the following scenario: An auditor wishes to verify whether a model utilized by an insurance company to justify claim decisions (approve/deny claim) is accurate on a dataset of the auditor's choosing. At the same time, the company does not want to reveal its model due to concerns about privacy and business competition. Certifiable ML works use cryptographic techniques to reconcile these seemingly conflicting goals.

**Zero-knowledge proofs** Among these techniques, the central tool is *zero-knowledge (ZK) proofs*, a classical cryptographic primitive, which allows one party (a *prover*) prove a statement x to another party (*verifier*) without revealing anything else apart from the validity of this statement. Such proofs are constructed for a concrete NP relation $\mathcal{R}$, which is used to formalize what it means for a statement to be true by specifying the type of evidence (witness w) that certifies it. The statement x is public, the witness w is private, and the zk proof checks $(x, w) \in \mathcal{R}$, without revealing w. In certifiable ML, such proofs allow model provider (prover) to formally prove that a model (witness) satisfies a desired property (e.g., accuracy, fairness, or inference correctness) on a given test dataset (statement) without learning anything else about the model or the training data. More formally:

**Definition 1** (Proof System). *An (interactive) proof system* ZKP *for an NP relation $\mathcal{R}$ is a tuple of interactive Turing machines $(\mathcal{P}, \mathcal{V})$, where $\mathcal{P}$ is prover and $\mathcal{V}$ is verifier. Let $b \leftarrow \langle \mathcal{P}(w), \mathcal{V} \rangle (x)$ denote the interaction between $\mathcal{P}$ and $\mathcal{V}$, where both $\mathcal{P}$ and $\mathcal{V}$ take x as common inputs, and $\mathcal{P}$ additionally takes w as a private input. At the end of interaction, $\mathcal{V}$ halts by outputting a binary b.*

Proof systems that are used in ML auditing typically require the following security properties: For an NP relation $\mathcal{R}$, they must provide *completeness* (i.e., if prover and verifier follow the protocol with input $(x, w) \in \mathcal{R}$, verifier always accepts), *(knowledge) soundness* (i.e., if verifier accepts, then it must be that prover owns a valid witness w satisfying given NP relation w.r.t. statement x), and *zero knowledge* (i.e., the transcript of the interaction between the prover and the (malicious) verifier leaks nothing except that there exists a witness w such that $(x, w) \in \mathcal{R}$). See §A.4 for formal definitions and §G for an overview of the NP relations underlying common zk proofs in certifiable ML (e.g., proofs of training, inference, etc.).

Returning to our example, suppose the insurance company has successfully passed an audit and can now deploy its model. How can a customer submitting inference queries be assured that the company continues to use the *certified* model—rather than switching to a different, unverified one? Again, the company still wishes to keep its model private.

**Cryptographic Commitment Schemes** The standard cryptographic tool here is *commitments*, which bind the provider to a single private model during the audit. This prevents "model switching" and ensures that model used in deployment is the same as the one that was certified:

**Definition 2** (Commitment Scheme). *A commitment scheme is an algorithm* Commit, *which is executed as* com $\leftarrow$ Commit$(m; \rho)$. *It takes as input a message $m \in \{0, 1\}^{\ell_m(\lambda)}$, a uniformly*

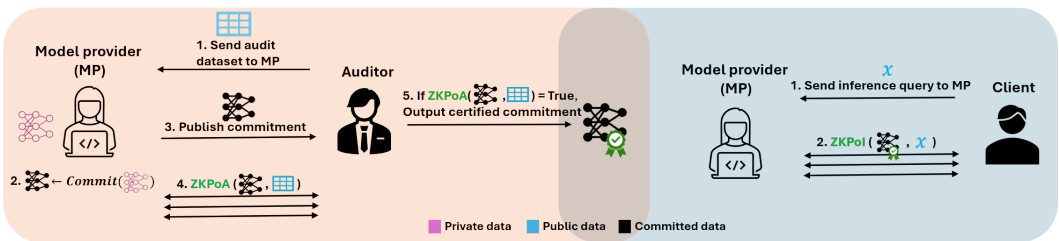

Figure 1: Simplified protocol flow for (insecure) ZK-based ML certification. Left: The model provider, after observing the audit dataset, commits to a model and engages with the auditor in a zero-knowledge proof of accuracy (ZKPoA). If the audit succeeds, the auditor certifies the committed model. Right: For each new inference query, the model provider interacts with the client in a zero-knowledge proof of inference (ZKPoI) protocol, ensuring that the result is consistent with the previously certified commitment.

*sampled randomness $\rho \in \{0,1\}^{\ell_r(\lambda)}$, and returns a commitment* com $\in \{0,1\}^{\ell_c(\lambda)}$. *Here $\ell_m, \ell_r, \ell_c$ are some polynomials in $\lambda$, the security parameter (determining the desired level of security).*

We require two security properties: *hiding* (i.e., given a commitment com, it leaks nothing about the message $m$), and *binding* (i.e., it is computationally infeasible to find two different pairs $(m, \rho)$ and $(m', \rho')$ such that $\mathsf{Commit}(m; \rho) = \mathsf{Commit}(m'; \rho')$). See §A.5 for formal definitions.

Now, auditing may require publishing such a commitment to the model,[2] after which the client and insurance company engage in a ZK proof of inference against it. Figure 1 shows the full certification workflow, where the audit dataset is revealed to the model provider prior to committing to the model.

**Unifying Syntax for Prior Works** We will next discuss the security guarantees of works that address the first stage of certification—namely, proofs of accuracy, fairness, etc., between auditor and model provider. To analyze these systematically, rather than case by case, we abstract away implementation details and introduce a unifying syntax that captures a broad class of existing audit systems.

Given a predicate $f(h, S_{train}, S_{audit})$ and a distribution $\mathcal{D}$, we define the auditing scheme as follows:

1. Auditor samples $S_{\text{audit}} \sim \mathcal{D}$ (or uses a public one) and sends $S_{\text{audit}}$ to the model owner

2. Model owner sends cryptographic commitments to its model $\mathsf{com}_h \leftarrow \mathsf{Commit}(h)$ and to the training data $\mathsf{com}_{\text{train}} \leftarrow \mathsf{Commit}(S_{\text{train}})$ to the auditor

3. They interact to execute ZKP: $b \leftarrow \langle \mathcal{P}(h, S_{\text{train}}), \mathcal{V} \rangle (\mathsf{com}_h, \mathsf{com}_{\text{train}}, S_{\text{audit}})$, where Model owner plays $\mathcal{P}$ and the auditor plays $\mathcal{V}$ and outputs $b$.

If the output is 1, the auditor is convinced that $f(h, S_{train}, S_{audit}) = 1$, where $h$ and $S_{train}$ are the model and training data committed in $\mathsf{com}_h$ and $\mathsf{com}_{train}$. Depending on $f$, some steps may be omitted; e.g., for an audit that checks accuracy or demographic parity on $S_{\text{audit}}$, $\mathsf{com}_{train}$ is unnecessary (see examples of $f$ in § A.1). Further, in some works, e.g., Shamsabadi et al. (2022), the model owner, rather than the auditor, samples the audit dataset.

## 4 ATTACKING ML CERTIFICATION

Returning to our example, suppose the insurance company saves costs by *denying* claims. Intuitively, an accuracy audit with provable guarantees—such as those provided by zk proof-based systems—and with a sufficiently high threshold (e.g., passing only if accuracy on the auditor's dataset exceeds 95%) should prevent the company from deploying a model that unjustifiably denies too many claims.

We show that this intuition is false. Because machine learning is inherently data-dependent, certified properties need not hold once the model is deployed and applied to fresh data, even when drawn from the same distribution. More formally, while prior works certify that

$$f(h, S_{\text{train}}, S_{\text{audit}}) = 1$$

---

[2]The commitment may be signed by the auditor.

for some predicate $f$, a given model $h$, training data $S_{\text{train}}$, and audit dataset $S_{\text{audit}}$, this does not imply that the stronger property $F$ such that

$$F(h, S_{\text{train}}) = 1 \iff \Pr_{S_{\text{test}} \leftarrow \mathcal{D}}[f(h, S_{\text{train}}, S_{\text{test}}) = 1] > \mathsf{p}$$

where $\mathsf{p}$ is a non-negligible probability and $\mathcal{D}$ is a distribution over the entire population $Q = \{(x_i, y_i)\}_{i=1}^m$. The true goal of an audit, however, is precisely such stronger guarantees: an auditor typically seeks to ensure that a model remains fair, accurate, or robust not only on a particular dataset, but also on the unseen datasets it will encounter during deployment.

We show that this gap can be exploited. In particular, if $S_{\text{audit}}$ is known to the model provider before it is required to cryptographically commit to the model, the provider can ensure $f(h, S_{\text{train}}, S_{\text{audit}}) = 1$ (and thus pass the audit), *without* additionally satisfying $F$, which is the actual intended security property. A malicious model provider has strong incentives to do so: for example, the insurance company could deploy a model that maximizes accuracy on the audit dataset (and thus passes the audit), yet still unjustifiably denies numerous insurance claims.

**Attack Game with Known Audit Data** Before providing a concrete attack example, we introduce a theoretical tool – an attack game – which showcases the gap between verifying $f(h, S_{train}, S_{audit})$ (which is what prior approaches certified) and $F(h) = (\Pr_{S_{test} \leftarrow D}[f(h, S_{train}, S_{test}) = 1] > \mathsf{p})$ (the intuitive property that one would want to ensure) for audit schemes where the model owner is given the audit dataset at the beginning of the audit process.

For simplicity, we will assume that the audit process verifying $f(h, S_{train}, S_{audit})$ is perfectly secure, i.e., the outcome of $\langle \mathcal{P}(h, S_{\text{train}}), \mathcal{V} \rangle(\mathsf{com}_h, \mathsf{com}_{\text{train}}, S_{\text{audit}})$, where $\mathsf{com}_{train}$ is a commitment to $S_{train}$ and $\mathsf{com}_h$ is a commitment to $h$, is 1 if and only if $f(h, S_{train}, S_{audit}) = 1$.

In the game, the adversary will win only if it can come up with a model $h$ and training data $S_{train}$, such that: **(1)** $f(h, S_{train}, S_{audit}) = 1$, i.e, the adversary would pass an audit on the dataset $S_{audit}$, and **(2)** $F(h, S_{train}) = 0$. To make the attack even stronger, we require the adversary to additionally satisfy a utility requirement (formalized via a predicate $L$) in order to win the game. Intuitively, the goal of $L$ is to capture the actual intent of the malicious model owner: For example, in case of the insurance company that wishes to deny claims, we could use $L(h) = \Pr_{x \sim \{0,1\}^d}[h(x) = 0] > 0.9$.

**Definition 3** (Adaptive Training with Known Auditing Data). *Let $f : \{0, 1\}^* \times \{0, 1\}^* \times \{0, 1\}^* \to \{0, 1\}$ be a predicate verified by the model certification, and let $F : \{0, 1\}^* \times \{0, 1\}^* \to \{0, 1\}$ be the actual intended security property. Let $\mathcal{X}$ be the feature space and $\mathcal{D}$ be a distribution over $\mathcal{X}$. Let $L$ denote the utility predicate[3]. Consider the following game played by an adversary $\mathcal{A}$:*

1. *Sample $S_{audit} \sim \mathcal{D}$*

2. *Given $S_{audit}$, $\mathcal{A}$ outputs a hypothesis $h_A$ and a training dataset $S_{train}$*

3. *Obtain $b = f(h_A, S_{train}, S_{audit})$*

4. *The output of the game is 1 ($\mathcal{A}$ 'wins') iff $b = 1$, $F(h_A, S_{train}) = 0$, and $L(h_A, S_{train}) = 1$.*

   *The output is 0 ($\mathcal{A}$ 'loses') otherwise.*

Looking ahead, our security definition provides a (relaxed) guarantee that $F(h, S_{train}) = 1$, hence if an adversary wins the attack game, the corresponding audit scheme cannot be secure under the definition in §5. We also note that proving security (§5) does not require knowledge of the utility predicate $L(h)$; this predicate strengthens our attack examples by capturing additional objectives a malicious model provider may pursue beyond violating the audit guarantees.

## 4.1 Example of a Data Forging Attack

We now give a concrete example of an attack within the framework of Def. 3 for the proofs of accuracy (e.g., (Zhang et al., 2020)) which utilize a dataset known to the model provider.

---

[3]We assume that distribution $\mathcal{D}$ is implicitly "known" to $L$ (it is either hard-coded or provided as a parameter to $\mathcal{L}$). For simplicity of notation, we omit $\mathcal{D}$ from the description of $\mathcal{L}$.

We consider our running example of an insurance company audit. Say the company uses a *decision tree* model (see §A.3 for background). The auditor wishes to check that the model is highly accurate, i.e., $F(h) = \Pr_{x \sim \{0,1\}^d}[h(x) = y_x] > 0.95$, where $y_x$ is the true label of $x$. The insurance company saves on costs when it denies claims, i.e., the model provider's utility $L(h) = (\Pr_{x \sim \{0,1\}^d}[h(x) = 0] > 0.9)$.[4] Following Def. 3, given audit dataset $S_{audit}$, we wish to construct a hypothesis $h_A$ that passes the audit, while simultaneously achieving $F(h_A, S_{train}) = 0$ and $L(h_A, S_{train}) = 1$.

**Attack Outline** At a high level, we obtain $h_A$ from *honestly* training a model on a dataset $S'_{train}$ crafted with the knowledge of $S_{audit}$ in mind. In more detail, we build $S'_{train}$ from the audit dataset $S_{audit}$ using the corresponding true labels—to ensure the model passes the accuracy test on $S_{audit}$—and augment it with points in the neighborhood of those audit points. The points in the neighborhood are labeled 0, so that outside the audit samples the model learns to label every point with a 0. More formally, to construct $S'_{train}$, for every point $r \in S_{audit}$ and every (numeric) feature $i$, we add $r + \varepsilon \vec{b}_i$ with label 0, where $\varepsilon$ is some small number and $\vec{b}_i$ is the $i$th basis vector. We also add every $r \in S_{audit}$ to $S'_{train}$ with honest labels (see Algorithm 1 for formal description). Then, we train a tree on $S'_{train}$ via a standard DT training algorithm **until leaves are homogeneous**.

---

**Algorithm 1** Data Forging Attack

---

**Input**: Audit set $S_{audit}$, dimension $d$, $\varepsilon > 0$
**Output**: Training data $S'_{train}$
   **function** ATTACK($S_{audit}, d, \varepsilon, g$)
      $S'_{train} \leftarrow S_{audit}$
      **for** $r \in S_{audit}$ **do**
         **for** $i \in [d]$ **do**
            $r_0 \leftarrow (r + \varepsilon \vec{b}_i, 0)$            ▷ $\vec{b}_i$ is the one-hot vector in dimension $i$
            $r_1 \leftarrow (r - \varepsilon \vec{b}_i, 0)$
            $S'_{train} \leftarrow S'_{train} \cup \{r_0, r_1\}$
      **return** $S'_{train}$

---

As we confirm in Fig. 5, this attack achieves good results. We now show that this attack allows the adversary to win in the attack game of Def 3. To this end, we first state the following theorem:

**Theorem 1.** *Classical decision tree training until homogeneity on the output of Algorithm 1 yields a tree $\mathcal{T}$ such that for every $x \in \mathbb{R}^d$, $\mathcal{T}(x) = 1$ only if $||x - r||_\infty < \varepsilon$ for some $r \in S_{audit}$.*

At a high level, the proof shows that if two points land in the same leaf, then any point lying between them on one coordinate must also fall in that leaf. Further, as the only non-zero points in $S'_{train}$ are audit points, every non-zero-labeled leaf contains an audit point. For any $x$ at least $\varepsilon$ away from all audit points, if $\mathcal{T}(x) = 1$, one can construct a nearby training point with label 0 that must lie in the same leaf, giving a contradiction as we trained until homogeneity. See §B.1 for details.   □

Thus, whenever a model provider generates a training dataset using Algorithm 1, an honestly trained decision tree that grows until homogeneity will achieve perfect accuracy on the audit dataset, yet predict zero for all inputs that lie outside an $\varepsilon$-neighborhood of the audit dataset points. Thus, for an appropriate choice of epsilon, the adversary wins in the game specified in Definition 3 with probability one.[5]

**Detection** Attacks by malicious model providers can typically be prevented by requiring cryptographic proofs that a model has been trained using a specific (benign) algorithm. However, such proofs are useless here: The model provider trains the model *honestly* using a standard training algorithm, and it's the training data itself that enables the attack. One might expect training data manipulation to be caught by statistical tests, e.g., Welch's $t$-test, which assess whether two samples are drawn from the same distribution. As we show in §C, however, this is not the case. We can cause the distributional properties of the training data to converge towards the audit data without sacrificing the efficacy of this attack by adding more copies of the audit data to the training data. This causes the audit data and the training data to appear as if they were drawn from the same distribution under a variety of statistical tests without impacting the model's ability to learn the desired behavior.

---

[4]For simplicity, we consider datapoints in $\{0, 1\}^d$
[5]Assuming that a model which almost always outputs 0 is not highly accurate in our scenario.

# 5 PROVABLY SECURE AUDITING PROTOCOLS

In this section, we provide an overview of our positive results, which are detailed in § E. We begin by defining the syntax of a secure auditing protocol.

**Definition 4** (Auditing Protocol). *An auditing protocol $\Pi$ for a predicate $F$ is a tuple of algorithms* $(\mathsf{Commit}, \mathsf{Prove}, \mathsf{Audit})$: *a commitment, a proving, and an auditing algorithm. Let* $(\mathsf{com}, b) \leftarrow \langle \mathsf{Prove}(h, S_{train}), \mathsf{Audit} \rangle$ *denote the interaction between* $\mathsf{Prove}$ *and* $\mathsf{Audit}$, *where* $\mathsf{Prove}$ *takes a hypothesis $h$ and optionally a training dataset $S_{train}$ as private input and outputs a commitment* $\mathsf{com}$ *during an execution, and* $\mathsf{Audit}$ *halts by outputting a binary $b$.*

**Framework of Provably Secure Auditing Protocols.** We define a *commit-sample-prove* auditing protocol $\Pi_{\mathrm{csp}} = (\mathsf{Commit}, \mathsf{Prove}, \mathsf{Audit})$ using an emprical predicate $f$ and a distribution $\mathcal{D}$ over a query space $Q = \{(x_i, y_i)\}_{i=1}^m$. Let $\mathsf{Commit}$ be a binding commitment scheme (§ A.5) and $\mathsf{ZKP} = (\mathcal{P}, \mathcal{V})$ be a ZK proof system for the following relation $\mathcal{R}$: for a pair of public statement $\mathsf{x} = (\mathsf{com}, S_{\mathrm{audit}})$ and private witness $\mathsf{w} = (h, \rho)$, we have $(\mathsf{x}, \mathsf{w}) \in \mathcal{R} \iff f(h, S_{\mathrm{audit}}) = 1 \land \mathsf{com} = \mathsf{Commit}(h; \rho)$. That is, the ZK proof ensures that the model $h$ committed in $\mathsf{com}$ satisfies the empirical predicate $f$ on the audit dataset $S_{\mathrm{audit}}$. While we focus on a protocol checking $F$ on a hypothesis $h$ only, the construction below can be naturally extended to a more complex $F$ and $f$ that additionally take a training dataset $S_{\mathrm{train}}$ as input.

$\langle \mathsf{Prove}(h), \mathsf{Audit} \rangle$

1. $\mathsf{Prove}$ computes $\mathsf{com} = \mathsf{Commit}(h; \rho)$ using a uniformly random string $\rho$ and sends $\mathsf{com}$ to $\mathsf{Audit}$.

2. $\mathsf{Audit}$ samples $S_{\mathrm{audit}} \leftarrow \mathcal{D}^n$ and sends it to $\mathsf{Prove}$.

3. $\mathsf{Prove}$ and $\mathsf{Audit}$ execute $b \leftarrow \langle \mathcal{P}(\mathsf{w}), \mathcal{V} \rangle(\mathsf{x})$, where $\mathsf{x} = (\mathsf{com}, S_{\mathrm{audit}})$ and $\mathsf{w} = (h, \rho)$. Here, $\mathsf{Prove}$ plays $\mathcal{P}$ and $\mathsf{Audit}$ plays $\mathcal{V}$.

4. $\mathsf{Prove}$ outputs $\mathsf{com}$, while $\mathsf{Audit}$ outputs $b$.

The key takeaway is that the audit dataset $S_{\mathrm{audit}}$ should be chosen *independently* of the model $h$ and the training dataset $S_{\mathrm{train}}$. Sampling $S_{\mathrm{audit}}$ after $h$ is committed ensures independence, rendering our earlier attack ineffective. The following theorem (formally stated and proved in § E.2) states that $\Pi_{\mathrm{csp}}$ satisfies the desired security properties (formally defined in § E.1) in a general fashion. Essentially, our result allows the protocol designer to choose an empirical predicate $f$ that approximates the desired property $F$ well enough, and then plug in any commitment scheme (§ A.5) and ZK proof system (§ A.4) satisfying the required security properties (§ A.4,§ A.5) to get a secure auditing scheme. We provide example instantiations of $f$ and $F$ for accuracy (§ E.2.1) and demographic parity auditing (§ E.2.2), both of which enable negligibly small false positive and negative rates with a sufficient number of samples $n$. Completeness, binding, and zero knowledge in the theorem follow directly from the properties of the underlying primitives, but knowledge soundness requires more care. In particular, the predicate $\tilde{F}$ ensured by auditing is a *relaxed version* of the original $F$ (and is tunable to control false positives). This reflects that $f$ is evaluated on a *finite sample*, which may not perfectly represent the property $F$ on the *underlying distribution*.

**Theorem 3 (informal).** *Let the empirical predicate $f$, the model predicate $F$, and the relaxed model predicate $\tilde{F}$ satisfy the following false negative and false positive rate bounds for every model $h$:*

$$\Pr_{S_{audit} \leftarrow \mathcal{D}^n}[f(h, S_{audit}) \neq 1 \mid F(h) = 1] \leq p_{fnr} \qquad \Pr_{S_{audit} \leftarrow \mathcal{D}^n}[f(h, S_{audit}) = 1 \mid \tilde{F}(h) \neq 1] \leq p_{fpr}$$

*If* $\mathsf{Commit}$ *and* $\mathsf{ZKP}$ *satisfy the standard security properties (§ A.4-A.5), then $\Pi_{csp}$ is a provably secure auditing protocol satisfying the following:*

***Completeness.*** *If an honest prover holds a model and a training set $h$, which satisfy property $F$, then this prover should pass the audit for $h$ (i.e,* $\mathsf{Audit}$ *outputs $b = 1$) except with probability $p_{fnr}$.*

***Binding.*** *No prover can change its model $h$ after committing to it.*

***Zero Knowledge.*** *An honest execution of the protocol between* $\mathsf{Prove}$ *and* $\mathsf{Audit}$ *does not reveal any information about the model $h$ beyond the fact that $F(h) = 1$.*

***$\tilde{F}$-relaxed Knowledge Soundness.*** *If a (potentially dishonest prover)* $\mathsf{Prove}^*$ *holds an invalid model $h$, i.e., $\tilde{F}(h) \neq 1$,* $\mathsf{Audit}$ *should detect it by outputting $b = 0$ except with probability $\approx p_{fpr}$.*

## 6 CASE STUDIES: VULNERABILITY TO DATA FORGING IN PREVIOUS WORK

Our formalization from §5 and §4 lets us test whether a protocol is vulnerable to data-forging. For works which reveal neither the model nor the training data, the check boils down to whether the prover is required to commit to the training data and/or to the model before seeing the audit dataset. We examined several prior works (Zhang et al., 2020; Shamsabadi et al., 2022; Yadav et al., 2024; Liu et al., 2021; Franzese et al., 2024; Shamsabadi et al., 2024; Kang et al., 2022; Wang and Hoang, 2023; Bourrée et al., 2025) with formal security guarantees. Surprisingly, the majority of the works either do not explicitly state when the audit dataset is revealed, or consider settings where the prover's training dataset and/or the model itself are assumed to be trusted (and are susceptible to data forging if the prover is actually malicious). Works that do not discuss the timing of the commitment often point out that their solution can be used to conduct audits using publicly known datasets, in which case the public dataset can be assumed to be known to the adversary prior to auditing, making the solution vulnerable to data-forging. We present our case studies in §D, where we examine the security models and techniques employed in each of the works, and discuss why a given approach is or is not vulnerable to data-forging attacks. We note that works that are not susceptible to data-forging attacks nonetheless provide only dataset-specific guarantees, i.e., their proofs certify properties solely on the chosen audit set/inference queries already submitted by clients, without extending to the underlying data distribution. It would be interesting to perform an analysis similar to that in §E.2.1 to derive formal guarantees that hold for the corresponding distributions. We summarize the results of our findings in Table 1.

## 7 EVALUATION

In this section we underscore the importance of data forging attacks by mounting proof of concept attacks for models trained on a variety of datasets. We show that our attack is effective in making inaccurate models appear accurate and unfair models appear fair, and empirically demonstrate a variety of other qualities, e.g. undetectability with a variety of statistical tools.

**Experimental Setup.** We use six well-known fairness benchmarking datasets in our experiments: ACSEmployment Ding et al. (2021), Adult Becker and Kohavi (1996), COMPAS Angwin et al. (2016), German Credit Hofmann (1994), Default Credit Yeh and hui Lien (2009), and Communities & Crime Redmond (2009). We implemented a modified version of our attack from §4.1 in Python 3.12.3 using SciKit-Learn version 1.6.1 that attempts to minimize an objective when deployed and evaluated its performance against a variety of datasets. For a given run, we split the dataset into an evaluation dataset consisting of $30\%$ of the data, an audit dataset containing 1000 data points, and an initial training data set. We represent the interpolation between a fully honest training run and a fully malicious one by the *attack parameter*, which takes a value between 0 (fully honest) and 1 (fully malicious). The attack parameter controls what proportion of audit data points are included in the training data and what proportion of the initial training data is labeled maliciously. The specifics of how honest and malicious data points are labeled depends on the objective.

To attack accuracy, we constructed a training dataset using a modified Algorithm 1, labeling the additional data from the attack with $1 - r_y$ rather than 0 and adding them to the initial training data set. For our fairness attack, we constructed the training data similarly, changing how honest and malicious data points were labeled. Honest data points were given random labels, while malicious data points were labeled according to their sensitive attribute. Both attacks then fit a decision tree to their constructed training data using SciKit-Learn's decision tree classifier class.

To ensure that our attack would evade statistical detection, we added extra copies of the audit data to the training data, as computed in Corollary 1 to pass Welch's $t$-test with significance level $0.05$.

**Attacking Accuracy Audits.** We ran our attack on six benchmark datasets – three shown in Figure 2 (remaining in Figure 4). Across all datasets, our data forging attack enforces high audit accuracy while simultaneously encouraging low performance on real-world evaluation data. Thus **our attack successfully makes inaccurate models *appear* accurate to an auditor.**

**Attacking Fairness Audits.** We also performed the attack while targeting demographic parity (using sex as the sensitive attribute) on three datasets, which we present in Figure 3. We were able to reliably

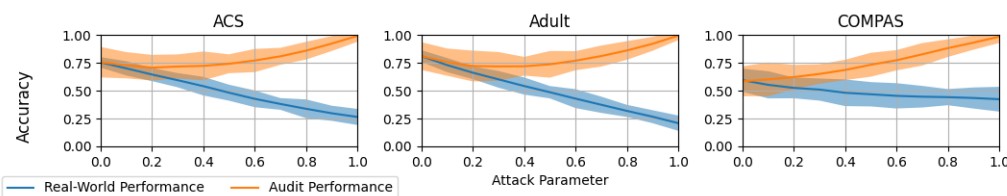

Figure 2: Performance of models trained on datasets constructed to minimize real-world accuracy while still passing an audit for several benchmarks. Values are averages over ten runs, error bars represent one standard deviation.

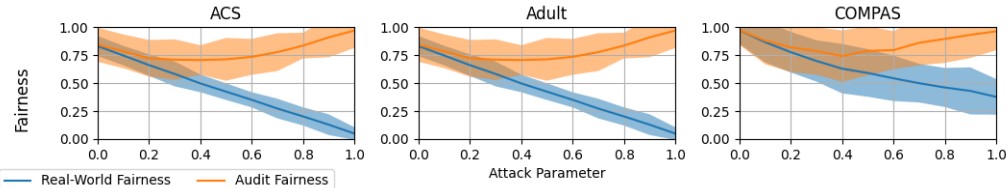

Figure 3: Fairness of models trained on constructed datasets using various benchmarks to target demographic parity. Values are averages over ten runs, error bars represent one standard deviation. Fairness means $1 -$ fairness gap.

train a model with close to $0$ fairness gap on the audit dataset, but close to $1$ fairness gap when deployed. In other words, **our attack successfully makes unfair models *appear* fair to an auditor.**

**Evading Detection via Statistical Methods.** We show how our attack can be executed in ways that evade detection by a variety of statistical approaches in Appendix Table 2. We were able to construct malicious training datasets with summary statistics that match those of the audit dataset very closely, and Welch's $t$-test and Levene's test regularly concluded that the audit and test datasets were drawn from the same distribution. This is consistent with our theoretical results in Appendix C.

**Additional Results.** An adversary can use data forging attacks to achieve concrete goals beyond degradation of accuracy or fairness, as we show in the Appendix H. For example, Figure 5 shows how an insurance provider could use our attack to hide the claim denial rate of a model from auditors. Figure 6 also shows preliminary results which suggest that our attack generalizes to neural networks.

## 8 DISCUSSION AND FUTURE WORK

This work brings attention to data-dependent vulnerabilities in cryptographic auditing methods for machine learning models. We propose an attack strategy that passes cryptographic certification while undermining the goals of those certifications for real-world performance. We then introduce new formal security definitions which address these vulnerabilities.

The attack strategy presented in this work poses several open questions. While we demonstrate the data forging attack is undetected even in the presence of Welch's $t$-test and Levene's test, it remains to be seen whether other statistical tests could effectively detect the attack. Based on the results that we have derived, we find it unlikely that other statistical tests will be effective in detecting the attack. However, we reserve such analysis for future work. We provide rigorous formal proofs that our attacks are effective on decision trees, and preliminary evidence that a similar approach generalizes to neural networks. Characterizing a formal relationship between neural network model capacity and attack effectiveness could be a promising direction in future work.

Our secure auditing template underscores the importance of keeping audit data hidden until the service provider's model is committed. This imposes a limitation on auditing in practice: auditors must either regularly gather fresh data (since the audit dataset is typically revealed during the audit), use additional cryptographic techniques such as secure multiparty computation to keep data hidden during the audit, or perform continuous auditing on user data. Each of these options has strengths and drawbacks which should be evaluated in more detail by future work.

## 9 ETHICS STATEMENT

Our work concerns techniques for cryptographic auditing of machine learning models. It is the hope of the authors that the insights in this paper can be used to improve cryptographic auditing techniques in order to make machine learning systems more adherent to ethical standards. We note that cryptographic auditing should not be considered a sufficient tool on its own for determining whether models are deployed ethically – as the present work highlights, it is sometimes possible to fulfill the technical criteria of model certification in order to pass an audit while circumventing the intention with which the audit was designed. Rather, cryptographic certification should be seen as a tool complementary to human oversight, which enables more efficient use of resources while auditing.

**LLM Usage.** The authors used LLMs to polish writing, and for assistance with literature search in some components of this paper. We also used generative AI to create some of the icons in Figure 1. In addition, we used an LLM for assistance with Lemma 2. We checked the proof assistance thoroughly by hand before including it in the paper.

## 10 REPRODUCIBILITY STATEMENT

We describe the experiments referenced in this work in §7 and §H, with reference to other parts of the paper which detail the specifics of the algorithms we employed. We will publish the source code used to run these experiments with the camera-ready version of the paper. All of our theoretical results are stated formally and proven in the Appendix sections §B, §C, and §E.

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

---

**Algorithm 2** Welch's $t$-test

---

**Input:** $\mathcal{X} = \{x_i\}_{i \in [n]}$, $\mathcal{Y} = \{y_i\}_{i \in [m]}$, where $x_i \sim X$ and $y_i \sim Y$, and a significance level $\alpha$
**Output**: Null hypothesis $H_0$ (i.e., $\mu_X = \mu_Y$) or alternative hypothesis $H_1$ (i.e., $\mu_X \neq \mu_Y$)

1: Compute sampled means $\bar{x} = \frac{\sum_i x_i}{n}$ and $\bar{y} = \frac{\sum_i y_i}{m}$

2: Compute sampled variances $v_x = \frac{\sum_i (\bar{x} - x_i)^2}{n-1}$ and $v_y = \frac{\sum_i (\bar{y} - y_i)^2}{m-1}$.

3: Compute the test statistic $t = \frac{\bar{x} - \bar{y}}{\sqrt{v_x/n + v_y/m}}$

4: Compute the degree of freedom $d = \frac{(g_x + g_y)^2}{g_x^2/(n-1) + g_y^2/(m-1)}$, where $g_x = v_x/n$ and $g_y = v_y/m$

5: Obtain the critical value $t_{cr}$ from the $t$-table, given $d$ and $\alpha$.

6: **If** $|t| < t_{cr}$ **return** $H_0$ **else return** $H_1$

---

## A  ADDITIONAL PRELIMINARIES

### A.1  EXAMPLE OF AUDITING PREDICATES

**Auditing Accuracy** To audit accuracy, we consider the empirical accuracy as follows:

$$\hat{\ell}_S(h) = \frac{1}{n} \sum_{(x,y) \in S} \mathbb{I}(h(x) \neq y)$$

where $n = |S|$, and define the empirical predicate $f$ as follows:

$$f(h, S_{\text{audit}}) = 1 \iff \hat{\ell}_S(h) \leq t$$

**Auditing Fairness with Demographic Parity** Demographic parity is one of the most basic fairness metrics, measuring the difference between the prediction probabilities conditioned on a sensitive attribute. We consider the empirical parity differences as follows:

$$\Delta_{\text{dp}}(h, S_{\text{audit}}) = \left| \frac{1}{n_0} \sum_{x \in S_0} \mathbb{I}(h(x) = 1) - \frac{1}{n_1} \sum_{x \in S_1} \mathbb{I}(h(x) = 1) \right|$$

where $s_x$ denotes the sensitive feature of a data point $x$, $S_0 = \{x \in S_{\text{audit}} : s_x = 0\}$, $S_1 = \{x \in S_{\text{audit}} : s_x = 1\}$, $n_0 = |S_0|$, and $n_1 = |S_1|$. To audit fairness w.r.t a model $h$ and a dataset $S_{\text{audit}}$, we define the corresponding empirical predicate $f$ as follows.

$$f(h, S_{\text{audit}}) = 1 \iff \Delta_{\text{dp}}(h, S_{\text{audit}}) \leq t$$

### A.2  WELCH'S $t$-TEST

**Welch's $t$-test** The goal of $t$-test is to determine whether the unknown population means of two groups are equal or not. That is, for random variables $X$ and $Y$, it compares the following hypotheses on their means $\mu_X = \mathrm{E}[X]$ and $\mu_Y = \mathrm{E}[Y]$:

- Null Hypothesis $H_0$: $\mu_X = \mu_Y$
- Alternative Hypothesis $H_1$: $\mu_X \neq \mu_Y$

Assuming that $X$ and $Y$ independently follow Gaussian distributions with unknown variances, Welch's $t$-test proceeds as in Algorithm 2.

### A.3  DECISION TREES

In our attack constructions we focus on decision tree models. Decision tree-based solutions are among the most popular machine learning algorithms, particularly known for their effectiveness in classification problems such as loan approval and fraud detection. A decision tree is trained by recursively partitioning the dataset from the root to the leaves. At each step, a split is determined by a splitting rule that aims to maximize an objective function, such as information gain. For prediction, the input follows a path from the root to a leaf, where at each internal node, the decision depends on whether the input satisfies the corresponding threshold (see Algorithm 3).

For completeness, in Algorithm 3 we present the algorithm for decision tree inference.

---

**Algorithm 3** Decision Tree Inference

---

**Input:** Decision tree $h$, input $\mathbf{a}$.
**Output:** Classification result.

1: Let cur $:= h.\text{root}$                                                   ▷ Set cur to be root of the tree
2: **while** cur is not a leaf **do**
3:      **if** $\mathbf{a}[\text{cur.attr}] < \text{cur.thr}$ **then**
4:         cur $:= \text{cur.left}$.                         ▷ Set cur to be current node's left child
5:      **else**
6:         cur $:= \text{cur.right}$.                       ▷ Set cur to be current node's right child
7: **return** cur.class

---

### A.4 SECURITY PROPERTIES OF ZERO-KNOWLEDGE PROOFS

Let $\mathsf{ZKP} = (\mathcal{P}, \mathcal{V})$ be an interactive proof system for a relation $\mathcal{R} = \bigcup_{\lambda \in \mathbb{N}} \mathcal{R}_\lambda$. In what follows, we denote by PPT *probabilistic polynomial time*.

**Completeness** ZKP is (perfectly) *complete* if for any $(\mathsf{x}, \mathsf{w})$ satisfying $\mathcal{R}$, it holds that:

$$\Pr[1 \leftarrow \langle \mathcal{P}(\mathsf{w}), \mathcal{V} \rangle(\mathsf{x})] = 1.$$

**Knowledge Soundness** ZKP is (adaptively) *knowledge sound* with knowledge error $\kappa$ if for any (stateful) PPT adversary $\mathcal{P}^* = (\mathcal{P}_0, \mathcal{P}_1)$, there exists an expected polynomial time extractor $\mathcal{E}$ such that the following holds:

$$p_{\text{ext}} \geq p_{\text{acc}} - \kappa$$

where

$$p_{\text{ext}} = \Pr\left[R_\lambda(\mathsf{x}, \mathsf{w}) = 1 \,:\, \mathsf{x} \leftarrow \mathcal{P}_0(1^\lambda); \mathsf{w} \leftarrow \mathcal{E}_{\mathcal{P}}(\mathsf{x})\right]$$

$$p_{\text{acc}} = \Pr\left[b = 1 \,:\, \mathsf{x} \leftarrow \mathcal{P}_0(1^\lambda); b \leftarrow \langle \mathcal{P}_1, \mathcal{V} \rangle(\mathsf{x})\right]$$

where $\mathcal{E}$ has non-black-box access to $\mathcal{P}^*$. Informally, this means that any cheating prover must know a valid witness if it convinces verifier.

**Zero-Knowledge** Let $\text{view}_{\mathcal{V}}^{\mathcal{P}(\mathsf{w})}(\mathsf{x})$ be a string consisting of all the incoming messages that $\mathcal{V}$ receives from $\mathcal{P}$ during the interaction $\langle \mathcal{P}(\mathsf{w}), \mathcal{V} \rangle(\mathsf{x})$, and $\mathcal{V}$'s random coins. $\Pi$ is (honest verifier) *zero-knowledge* if there exists a PPT simulator $\mathcal{S}$ such that for any adversary $\mathcal{A}$ and any $(\mathsf{x}, \mathsf{w}) \in \mathcal{R}_\lambda$, the following is negligible in $\lambda$.

$$\left| \Pr\left[b = 1 \,:\, b \leftarrow \mathcal{A}(\text{view}_{\mathcal{V}}^{\mathcal{P}(\mathsf{w})}(\mathsf{x}))\right] - \Pr\left[b = 1 \,:\, \text{view}' \leftarrow \mathcal{S}(\mathsf{x}); \quad b \leftarrow \mathcal{A}(\text{view}')\right] \right|$$

Informally, this means that the protocol execution reveals no information to $\mathcal{V}$ about $\mathsf{w}$.

### A.5 SECURITY PROPERTIES OF COMMITMENT SCHEMES

Let Commit be a commitment scheme. For simplicity, we omit the key generation algorithm Gen for simplicity and present a class of the simplest commitments whose openings are checked by re-computing and comparing (e.g., hash commitment $H(m||\rho)$). More generally, some commitment schemes require a separate verification algorithm Verify to check the validity of a commitment given some *decommitment* information. Our auditing framework can be extended to such schemes by having the model provider prove the knowledge of the decommitment information in zero knowledge.

**Binding** Commit is *computationally binding* if for any PPT adversary $\mathcal{A}$, the following is negligible in $\lambda$:

$$\Pr\left[\mathsf{Commit}(m; \rho) = \mathsf{Commit}(m'; \rho') \wedge m \neq m' \,:\, (m, m', \rho, \rho') \leftarrow \mathcal{A}(1^\lambda)\right]$$

**Hiding** Commit is *computationally hiding* if for any PPT adversary $\mathcal{A}$, the following is negligible in $\lambda$:

$$\left| \Pr\left[b = 1 \,:\, \begin{array}{l} m_0, m_1 \leftarrow \mathcal{A}(1^\lambda); \rho \leftarrow \{0,1\}^{\ell_r(\lambda)}; \\ \mathsf{com} \leftarrow \mathsf{Commit}(m_b; \rho); b \leftarrow \{0,1\}; b' \leftarrow \mathcal{A}(\mathsf{com}) \end{array}\right] - \frac{1}{2} \right|$$

## B    Deferred Proofs

### B.1    Proof of Theorem 1

*Proof.* First, let us show that if points $a$ and $b$, where $a_i < c_i < b_i$ and $c_j = b_j$ for all $j \neq i$, get sorted into the same decision tree leaf, then $c$ is also sorted into that same leaf. Notice that in order for two points $x$ and $y$ to get sorted into different leaves, there must be a node which splits on a feature $i$ such that $x_i \neq y_i$ and $x_i \leq t \leq y_i$ or $y_i \leq t \leq x_i$ where $t$ is the threshold to split upon. Then if $b$ and $c$ were sorted into separate leaves, there must be a node on the path that $b$ takes through the tree that splits on feature $i$ with a threshold $t$ that satisfies $c_i \leq t \leq b_i$. However, such a node would also sort $a$ distinctly from $b$, so such a node cannot occur. Thus, $b$ and $c$ must be sorted into the same leaf.

Now, note that as we train the tree until its leafs are homogeneous, every datapoint in $S'_{train}$ must be classified correctly (according to the label we assigned to it in Algorithm 1). Further, since the only datapoints in $S'_{train}$ with non-zero labels are datapoints from $S_{audit}$, for every leaf in $\mathcal{T}$ that is associated with a non-zero class, we have at least one $r \in S_{audit}$ that gets sorted into this leaf.

Consider $x \in \mathbb{R}^d$ such that $||x - r||_\infty \geq \varepsilon$ for all $r \in S_{audit}$. Say $\mathcal{T}(x) = 1$, i.e., there exists a leaf such that $x$ belongs to this leaf and the leaf corresponds to class one. Consider $r \in S_{audit}$ that belongs to this leaf (by above, such $r$ exists). By the definition of the L-infinity norm there exists some dimension $i$ where $|r_i - x_i| > \varepsilon$. Suppose $r_i - x_i > \varepsilon$. Notice that there is a point $r - \varepsilon \vec{b}_i \in S'_{train}$ which satisfies that $(r - \varepsilon \vec{b}_i)_j = r_j$ for all $j \neq i$, and where $x_i < (r - \varepsilon \vec{b}_i)_i < r_i$. Then by above, $r - \varepsilon \vec{b}_i$ must be sorted into the same leaf as $r$ and $x$. But $r - \varepsilon \vec{b}_i$ has label $g(r) = 0$, while for $x$ holds $\mathcal{T}(x) = 1$. Thus, we found a contradiction. The same argument holds if $x_i - r_i > \varepsilon$, but using the point $r + \varepsilon \vec{b}_i$ instead of $r - \varepsilon \vec{b}_i$. $\square$

## C    Attack Detection

While proof of training alone cannot detect the attack above (as it relies on training the decision tree entirely honestly), nor can a black-box audit where the model owner knows the audit data before training time, we might still hope to detect when these attacks occur. For example, we might hope to conduct statistical tests on the training data to determine if it was honestly sampled from the underlying distribution or if it was adversarially constructed. In such a case, we cannot directly compare the training data to the true distribution of real data because the underlying distribution is not fully known to the auditor. Instead, we must compare the training data with a sample from that distribution. In the most simple case, this sample is the reference set $S_{audit}$.

We argue that under a certain family of functions, our constructed training set is indistinguishable from $S_{audit}$.

**Definition 5.** *Suppose $\vec{\alpha}$ is a set of bins over $d$ dimensions. Then $H_{\vec{\alpha}} : (\mathbb{R}^d \times \{0,1\})^* \to \mathcal{H}$ is the function which takes databases over $d$ features and a binary classification to their normalized histogram with bins $\vec{\alpha}$.*

**Definition 6.** *A function $f : (\mathbb{R}^d \times \{0,1\})^* \to \mathbb{R}$ is called $(\gamma, c)$-magnitude insensitive if there exists a choice of bins $\vec{\alpha}$ and function $f' : \mathcal{H} \to \mathbb{R}$ such that $|f(D) - f'(H_{\vec{\alpha}}(D))| < \gamma$ for all $D \in (\mathbb{R}^d \times \{0,1\})^*$ and $|f'(H_{\vec{\alpha}}(D)) - f'(H_{\vec{\alpha}}(D||r))| \leq \frac{c}{|D|}$ for all $D \in (\mathbb{R}^d \times \{0,1\})^*$ and $r \in \mathbb{R}^d \times \{0,1\}$.*

**Theorem 2.** *If $f$ is $(\gamma, c)$-magnitude insensitive, then $\left| f(S_{audit}) - f\left( S_{audit}^k || \delta \right) \right| \leq \varepsilon$ for any $\varepsilon > 2\gamma$ and $k \geq \frac{2dc}{\varepsilon - 2\gamma}$, where $\delta$ is the additional training data created by Algorithm 1 when run with input $S_{audit}, d, \varepsilon, g$ for any $g$.*

*Proof.* We will write $f'$ to be the $\gamma$-approximation of $f$ guaranteed to exist by the fact that $f$ is $(\gamma, c)$-magnitude insensitive. Observe that because $H_{\vec{\alpha}}$ takes databases to their normalized histograms, $H_{\vec{\alpha}}(S_{audit}) = H_{\vec{\alpha}}\left( S_{audit}^k \right)$, because the non-normalized histograms of the two databases are simply scaled versions of one another.

Next, it will be helpful to show that for any two databases $D_1, D_2 \in (\mathbb{R}^d \times \{0,1\})^*$, we have $|f'(H_{\vec{\alpha}}(D_1)) - f'(H_{\vec{\alpha}}(D_1||D_2))| \leq c\frac{|D_2|}{|D_1|}$. Let us write $D_2 = d_1||d_2||\ldots||d_{|D_2|}$. Then we get that

$$
\begin{aligned}
&|f'(H_{\vec{\alpha}}(D_1)) - f'(H_{\vec{\alpha}}(D_1||D_2))| \\
&= |f'(H_{\vec{\alpha}}(D_1)) - f'(H_{\vec{\alpha}}(D_1||d_1)) + f'(H_{\vec{\alpha}}(D_1||d_1)) - \ldots \\
&\qquad + f'(H_{\vec{\alpha}}(D_1||d_1||d_2||\ldots||d_{|D_2|-1})) - f'(H_{\vec{\alpha}}(D_1||D_2))| \\
&\leq |f'(H_{\vec{\alpha}}(D_1)) - f'(H_{\vec{\alpha}}(D_1||d_1))| + |f'(H_{\vec{\alpha}}(D_1||d_1)) - f'(H_{\vec{\alpha}}(D_1||d_1||d_2)| + \ldots \\
&\qquad + |f'(H_{\vec{\alpha}}(D_1||d_1||d_2||\ldots||d_{|D_2|-1})) - f'(H_{\vec{\alpha}}(D_1||D_2))| \\
&\leq \frac{c}{|D_1|} + \frac{c}{|D_1|+1} + \ldots + \frac{c}{|D_1|+|D_2|-1} \\
&\leq c\frac{|D_2|}{|D_1|}
\end{aligned}
$$

Then we can apply this to $S_{audit}^k$ and $S_{audit}^k||\delta$; recall that $|\delta| = 2d|S_{audit}|$. Then we see that

$$
\begin{aligned}
\left|f'\left(H_{\vec{\alpha}}(S_{audit})\right) - f'\left(H_{\vec{\alpha}}\left(S_{audit}^k||\delta\right)\right)\right| &= \left|f'\left(H_{\vec{\alpha}}\left(S_{audit}^k\right)\right) - f'\left(H_{\vec{\alpha}}\left(S_{audit}^k||\delta\right)\right)\right| \\
&\leq c\frac{2d|S_{audit}|}{k|S_{audit}|} \\
&\leq c\frac{2d}{\left(\frac{2dc}{\varepsilon-2\gamma}\right)} = \varepsilon - 2\gamma
\end{aligned}
$$

We have two cases now.

Case 1: $f'\left(H_{\vec{\alpha}}(S_{audit})\right) \geq f'\left(H_{\vec{\alpha}}\left(S_{audit}^k||\delta\right)\right)$. Then we have

$$
\begin{aligned}
\varepsilon - 2\gamma &\geq f'\left(H_{\vec{\alpha}}(S_{audit})\right) - f'\left(H_{\vec{\alpha}}\left(S_{audit}^k||\delta\right)\right) \\
&= f(S_{audit}) - f(S_{audit}) + f'(H_{\vec{\alpha}}(S_{audit})) \\
&\qquad - f(S_{audit}^k||\delta) + f(S_{audit}^k||\delta) - f'(H_{\vec{\alpha}}(S_{audit}^k||\delta)) \\
&\geq f(S_{audit}) - |f(S_{audit}) - f'(H_{\vec{\alpha}}(S_{audit}))| \\
&\qquad - f(S_{audit}^k||\delta) - |f(S_{audit}^k||\delta) - f'(H_{\vec{\alpha}}(S_{audit}^k||\delta))| \\
&\geq f(S_{audit}) - \gamma - f(S_{audit}^k||\delta) - \gamma
\end{aligned}
$$

and so we see that $\varepsilon \geq f(S_{audit}) - f(S_{audit}^k||\delta)$. We also have

$$
\begin{aligned}
f(S_{audit}) - f(S_{audit}^k||\delta) &= f'(H_{\vec{\alpha}}(S_{audit})) - f'(H_{\vec{\alpha}}(S_{audit})) + f(S_{audit}) \\
&\qquad - f'(H_{\vec{\alpha}}(S_{audit}^k||\delta)) + f'(H_{\vec{\alpha}}(S_{audit}^k||\delta)) - f(S_{audit}^k||\delta) \\
&\geq f'(H_{\vec{\alpha}}(S_{audit})) - |f'(H_{\vec{\alpha}}(S_{audit})) - f(S_{audit})| \\
&\qquad - f'(H_{\vec{\alpha}}(S_{audit}^k||\delta)) - |f'(H_{\vec{\alpha}}(S_{audit}^k||\delta)) - f(S_{audit}^k||\delta)| \\
&\geq f'(H_{\vec{\alpha}}(S_{audit})) - \gamma - f'(H_{\vec{\alpha}}(S_{audit}^k||\delta)) - \gamma \\
&\geq -2\gamma \\
&> -\varepsilon
\end{aligned}
$$

Then $|f(S_{audit}) - f(S_{audit}^k||\delta)| \leq \varepsilon$.

Case 2: $f'(H_{\vec{\alpha}}(S_{audit})) \leq f'(H_{\vec{\alpha}}(S_{audit}^k||\delta))$. Then we have

$$
\begin{aligned}
\varepsilon - 2\gamma &\geq f'\left(H_{\vec{\alpha}}\left(S_{audit}^k||\delta\right)\right) - f'\left(H_{\vec{\alpha}}(S_{audit})\right) \\
&= f(S_{audit}^k||\delta) - f(S_{audit}^k||\delta) + f'(H_{\vec{\alpha}}(S_{audit}^k||\delta)) \\
&\qquad - f(S_{audit}) + f(S_{audit}) - f'(H_{\vec{\alpha}}(S_{audit})) \\
&\geq f(S_{audit}^k||\delta) - |f(S_{audit}^k||\delta) - f'(H_{\vec{\alpha}}(S_{audit}^k||\delta))| \\
&\qquad - f(S_{audit}) - |f(S_{audit}) - f'(H_{\vec{\alpha}}(S_{audit}))| \\
&\geq f(S_{audit}^k||\delta) - \gamma - f(S_{audit}) - \gamma
\end{aligned}
$$

and so we see that $\varepsilon \geq f(S_{audit}^k||\delta) - f(S_{audit})$. We also have

$$
\begin{aligned}
f(S_{audit}^k||\delta) - f(S_{audit}) &= f'(H_{\vec{\alpha}}(S_{audit}^k||\delta)) - f'(H_{\vec{\alpha}}(S_{audit}^k||\delta)) + f(S_{audit}^k||\delta) \\
&\quad - f'(H_{\vec{\alpha}}(S_{audit})) + f'(H_{\vec{\alpha}}(S_{audit})) - f(S_{audit}) \\
&\geq f'(H_{\vec{\alpha}}(S_{audit}^k||\delta)) - |f'(H_{\vec{\alpha}}(S_{audit}^k||\delta)) + f(S_{audit}^k||\delta)| \\
&\quad - f'(H_{\vec{\alpha}}(S_{audit})) - |f'(H_{\vec{\alpha}}(S_{audit})) - f(S_{audit})| \\
&\geq f'(H_{\vec{\alpha}}(S_{audit}^k||\delta)) - \gamma - f'(H_{\vec{\alpha}}(S_{audit})) - \gamma \\
&\geq -2\gamma \\
&\geq -\varepsilon
\end{aligned}
$$

Then $|f(S_{audit}) - f(S_{audit}^k||\delta)| \leq \varepsilon$. $\qquad\square$

This theorem does not suggest that it is completely impossible to detect the attack given in Algorithm 1. Rather, it only precludes detection by a certain class of functions. However, we argue that this class is expansive and covers many intuitive approaches.

The sole requirement for the audit metric $f$ is that it must be approximable by $f'$ which satisfies three properties. Firstly, $f'$ operates over histograms for some choice of bins $\vec{\alpha}$. This is a necessary condition, as if $f$ were not approximable by a function over a binning of the training data, we could drastically change the audit outcome by simply adding a small amount of noise to the data. Next, $f'$ must be relatively insensitive to additional data. The intuition here is that no individual datapoint should dramatically change the outcome of the audit. Finally, $f'$ operates over normalized histograms. This property is necessary for the proof to go through, but is satisfied by many intuitive audit metrics. For example, the mean and standard deviation of a feature (even conditioned on any arbitrary set of features) are approximable from a normalized histogram.

**Lemma 1.** *Let $\mu_j(D)$ be the mean of (bounded) feature $j$ of a dataset $D$. Then for every $\gamma > 0$, $\mu_j(D)$ is $(\gamma, M - m)$-magnitude insensitive, where $B$ is the set of bins in the histogram and $M, m$ are an upper and lower bound on possible $j$-values respectively.*

*Proof.* Notice that $\mu_j(D) \approx \sum_{i \in B} p_i x_{j,i}$ where $B$ is the set of bins in the histogram, $p_i$ is the height of bin $i$ in the normalized histogram of $D$, and $x_{j,i}$ is the $j$-value of bin $i$. Let us show that for any $\gamma > 0$, there exists a binning of the data such that this is a $\gamma$-approximation of $\mu_j(D)$. Let the bins in feature $j$ have width $\gamma$. Then for each datapoint $d$ with $j$ value $j_d$, bin $i$, and binned $j$-value $x_{j,i}$, we have that $|x_{j,i} - j_d| \leq \gamma$. Then

$$
\begin{aligned}
\sum_{i \in B} p_i x_{j,i} &= \sum_{i \in B} \frac{c_i}{|D|} x_{j,i} \\
&= \sum_{d \in D} \frac{1}{|D|} x_{j,i} \\
\implies \left| \sum_{i \in B} p_i x_{j,i} - \sum_{d \in D} \frac{1}{|D|} j_d \right| &= \left| \sum_{d \in D} \frac{1}{|D|} x_{j,i} - \sum_{d \in D} \frac{1}{|D|} j_d \right| \\
&= \left| \frac{1}{|D|} \sum_{d \in D} (x_{j,i} - j_d) \right| \\
&\leq \frac{1}{|D|} \sum_{d \in D} |x_{j,i} - j_d| \\
&\leq \frac{1}{|D|} \sum_{d \in D} \gamma \\
&= \gamma
\end{aligned}
$$

Next, let us show that the sensitivity of our approximation of $\mu_j$ is upper bounded by $\frac{M-m}{|D|}$. Notice that by adding a single point, one histogram bin will increase by 1 and the rest will be unchanged.

Then for every bin $k$,

$$\sum_{i \in B} \frac{c_i}{|D|+1} x_{j,i} + \frac{1}{|D|+1} x_{j,k} - \sum_{i \in B} \frac{c_i}{|D|} x_{j,i} = \sum_{i \in B} c_i x_{j,i} \left( \frac{1}{|D|+1} - \frac{1}{|D|} \right) + \frac{x_{j,k}}{|D|+1}$$

$$= - \left( \sum_{i \in B} \frac{c_i x_{j,i}}{|D|^2 + |D|} \right) + \frac{x_{j,k}}{|D|+1}$$

$$\leq - \left( \frac{m}{|D|+1} \right) + \frac{M}{|D|+1}$$

$$\leq \frac{M-m}{|D|}$$

$$\sum_{i \in B} \frac{c_j}{|D|+1} x_{j,i} + \frac{1}{|D|+1} x_{j,k} - \sum_{i \in B} \frac{c_j}{|D|} x_{j,i} = - \left( \sum_{i \in B} \frac{c_i x_{j,i}}{|D|^2 + |D|} \right) + \frac{x_{j,k}}{|D|+1}$$

$$\geq - \left( \frac{M}{|D|+1} \right) + \frac{m}{|D|+1}$$

$$\geq \frac{m-M}{|D|}$$

So we have that the sensitivity is no greater than $\frac{M-m}{|D|}$. $\square$

We will proceed to use this fact to show that Welch's $t$-test will fail to detect this attack.

**Corollary 1.** *Given an audit dataset $S_{audit}$ and significance level $\alpha$, we can use Algorithm 1 to construct a training dataset $S'_{train}$ such that for any feature $j$, $S'_{train}$ passes Welch's t-test when its values in feature $j$ are compared to those of $S_{audit}$ with significance level $\alpha$.*

Before we can prove this corollary, we will need a lemma which bounds the concentration of the Student's $t$-distribution.

**Lemma 2.** *If $X$ and $Z$ are random variables drawn independently from the Student's t-distribution with $\nu$ degrees of freedom and the standard normal distribution respectively, then for every $t > 0$, we have*

$$\Pr[|X| < t] \leq \Pr[|Z| < t]$$

*Proof.* We will write $F_X(t)$ to denote the CDF of random variable $X$ evaluated at $t$, and $f_X(t)$ the PDF. We will also write $\mathbb{E}_X(g(X))$ to be the expected value of $g(X)$ with randomness over $X$. Let us begin by demonstrating that for all $t < 0$, we have $F_X(t) > F_Z(t)$. First, recall that if $W$ and $Y$ are drawn from the $\chi^2$ distribution with $\nu$ degrees of freedom and the standard normal distribution respectively, then $Y\sqrt{\frac{\nu}{W}}$ is distributed according to the Student's $t$-distribution with $\nu$ degrees of freedom, so let us write $X = Y\sqrt{\frac{\nu}{W}}$. Then according to the law of total probability, we have

$$F_X(t) = \int_0^\infty F_Y \left( t\sqrt{\frac{w}{\nu}} \right) f(w) dw$$

$$= \mathbb{E}_W \left( F_Y \left( t\sqrt{\frac{W}{\nu}} \right) \right)$$

Notice that $\frac{d^2}{dt^2} F_Y(t) = \frac{d}{dt} f_Y(t) = \frac{d}{dt} \frac{1}{\sqrt{2\pi}} e^{-\frac{t^2}{2}} = -\frac{t}{\sqrt{2\pi}} e^{-\frac{t^2}{2}} > 0$ when $t < 0$. Then since $t\sqrt{\frac{W}{\nu}}$ must be less than 0, we can apply Jensen's inequality to get

$$F_X(t) = \mathbb{E}_W \left( F_Y \left( t\sqrt{\frac{W}{\nu}} \right) \right)$$

$$\geq F_Y \left( \mathbb{E}_W \left( t\sqrt{\frac{W}{\nu}} \right) \right)$$

$$= F_Y \left( t\mathbb{E}_W \left( \sqrt{\frac{W}{\nu}} \right) \right)$$

Then since $\frac{d^2}{du^2}\sqrt{u} = -\frac{1}{4\sqrt{u^3}} \leq 0$, we get that $\mathbb{E}_W\left(\sqrt{\frac{W}{\nu}}\right) \leq \sqrt{\frac{\mathbb{E}_W(W)}{\nu}} = \sqrt{\frac{\nu}{\nu}} = 1$. So because $t < 0$, we can see that $t\mathbb{E}_W\left(\sqrt{\frac{W}{\nu}}\right) \geq t$, and since $F_Y(u)$ is increasing, we get

$$F_X(t) \geq F_Y\left(t\mathbb{E}_W\left(\sqrt{\frac{W}{\nu}}\right)\right)$$
$$\geq F_Y(t)$$

Since $f_X$ and $f_Y$ are both symmetric about $t = 0$, it then follows by a symmetric argument that for all $t > 0$, $F_X(t) \leq F_Y(t)$. Then we see that for any $t > 0$,

$$\Pr[|X| < t] = F_X(t) - F_X(-t)$$
$$\leq F_Y(t) - F_Y(-t)$$
$$= \Pr[|Y| < t]$$
$$= \Pr[|Z| < t]$$

Because $Y$ and $Z$ are independently and identically distributed. $\qquad\square$

We are now ready to prove Corollary 1.

*Proof of Corollary 1.* A pair of datasets $D_1, D_2$ pass Welch's $t$-test on feature $j$ if

$$\frac{|\mu_j(D_1) - \mu_j(D_2)|}{\sqrt{\frac{\sigma_1^2}{|D_1|} + \frac{\sigma_2^2}{|D_2|}}} \leq T_{\alpha,\nu}$$

where $\alpha$ is the desired significance level, $\nu$ is the degrees of freedom in the datasets, and $T_{\alpha,\nu}$ is the unique value such that

$$\Pr_{x \sim t(\nu)}[|x| \geq T_{\alpha,\nu}] = \alpha$$

where $t(\nu)$ is the Student's $t$-distribution with $\nu$ degrees of freedom. In our case, the $t$-test compares the reference dataset $S_{audit}$ with the training dataset $S'_{train}$.

The value of $\nu$, and thus the value of $T_{\alpha,\nu}$, depends on the size of the datasets, with the threshold $T_{\alpha,\nu}$ decreasing as the datasets grow large. However, we will use Lemma 2 to give a lower bound for $T_{\alpha,\nu}$ which is constant with respect to $|S'_{train}|$. Then, we will show that by Lemma 1 and Theorem 2 we can use Algorithm 1 to construct a malicious training dataset $S'_{train}$ which maintains an arbitrarily small test statistic, and in particular, a dataset such that the test statistic is below the lower bound on the threshold.

First, let us establish a lower bound on $T_{\alpha,\nu}$. Let us define $T'_\alpha$ to be the unique positive value such that

$$\Pr_{Z \sim \mathcal{N}(0,1)}[|Z| \geq T'_\alpha] = \alpha$$

Then recall that Lemma 2 gives us that

$$\Pr_{X \sim t(\nu)}[|X| < T'_\alpha] \leq \Pr_{Z \sim \mathcal{N}(0,1)}[|Z| < T'_\alpha]$$

If we write $f_X$ and $f_Z$ to represent the probability density functions (PDFs) of $X$ and $Z$ respectively, then we get equivalently that

$$\int_{-T'_\alpha}^{T'_\alpha} f_X(u)du \leq \int_{-T'_\alpha}^{T'_\alpha} f_Z(u)du$$

Then we see that

$$\Pr_{Z\sim\mathcal{N}(0,1)}[|Z| \geq T'_\alpha] = \Pr_{X\sim t(\nu)}[|X| \geq T_{\alpha,\nu}]$$

$$\implies \int_{-T'_\alpha}^{T'_\alpha} f_Z(u)du = \int_{-T_{\alpha,\nu}}^{T_{\alpha,\nu}} f_X(u)du$$

$$= \int_{-T_{\alpha,\nu}}^{-T'_\alpha} f_X(u)du + \int_{-T'_\alpha}^{T'_\alpha} f_X(u)du + \int_{T'_\alpha}^{T_{\alpha,\nu}} f_X(u)du$$

$$\leq \int_{-T_{\alpha,\nu}}^{-T'_\alpha} f_X(u)du + \int_{-T'_\alpha}^{T'_\alpha} f_Z(u)du + \int_{T'_\alpha}^{T_{\alpha,\nu}} f_X(u)du$$

$$\implies 0 \leq \int_{-T_{\alpha,\nu}}^{-T'_\alpha} f_X(u)du + \int_{T'_\alpha}^{T_{\alpha,\nu}} f_X(u)du$$

Then because $f_X(x)$ is symmetric about $x = 0$, this yields

$$2\int_{T'_\alpha}^{T_{\alpha,\nu}} f_X(u)du \geq 0$$

and thus

$$\int_{T'_\alpha}^{T_{\alpha,\nu}} f_X(u)du \geq 0$$

Now recall the simple result from calculus that states that if $g$ is positive valued, then

$$\int_a^b g(x)dx \geq 0 \iff a \leq b$$

Then because $f_X$ is positive-valued, our prior result entails that $T_{\alpha,\nu} \geq T'_\alpha$, so $T'_\alpha$ is a lower bound on $T_{\alpha,\nu}$ that does not depend on $|S'_{train}|$.

Next, observe that the test statistic for Welch's $t$-test has the following upper bound:

$$\frac{|\mu_j(S'_{train}) - \mu_j(S_{audit})|}{\sqrt{\frac{\sigma^2_{train}}{|S'_{train}|} + \frac{\sigma^2_{audit}}{|S_{audit}|}}} \leq \frac{|\mu_j(S'_{train}) - \mu_j(S_{audit})|}{\sqrt{\frac{\sigma^2_{audit}}{|S_{audit}|}}}$$

Furthermore, Lemma 1 implies that for any $\varepsilon > 0$, we can choose $\gamma < \frac{\varepsilon}{2}$ such that $\mu_j$ is $(\gamma, c)$-magnitude insensitive, and so by Theorem 2, Algorithm 1 yields a dataset $S'_{train}$ such that $|\mu_j(S'_{train}) - \mu_j(S_{audit})| \leq \varepsilon$ when appropriately parameterized. Then let $\varepsilon = T'_\alpha \frac{\sigma_{audit}}{2\sqrt{|S_{audit}|}}$. This produces the result that

$$\frac{|\mu_j(S'_{train}) - \mu_j(S_{audit})|}{\sqrt{\frac{\sigma^2_{train}}{|S'_{train}|} + \frac{\sigma^2_{audit}}{|S_{audit}|}}} \leq \frac{2\varepsilon}{\sqrt{\frac{\sigma^2_{audit}}{|S_{audit}|}}}$$

$$= \frac{2}{\sqrt{\frac{\sigma^2_{audit}}{|S_{audit}|}}} T'_\alpha \frac{\sigma_{audit}}{2\sqrt{|S_{audit}|}}$$

$$= T'_\alpha$$

$$\leq T_{\alpha,\nu}$$

which passes the $t$-test for feature $j$. Finally, by choosing $k = \max_j \frac{4d(M_j - m_j)\sqrt{|S_{audit}|}}{T'_\alpha \sigma_{audit,j}}$ we get for every feature $i$ that $|\mu_i(S'_{train}) - \mu_i(S_{audit})| \leq 2\min_j T'_\alpha \frac{\sigma_{audit,j}}{2\sqrt{|S_{audit}|}} \leq 2T'_\alpha \frac{\sigma_{audit,i}}{2\sqrt{|S_{audit}|}}$, so $S'_{train}$ passes the $t$-test for feature $i$. $\qquad\square$

## D  CASE STUDY

We now discuss a number of state of the art works that consider the problem of privacy-preserving audititng. These works are focused on different auditing functions (accuracy, fairness, etc), different

types of machine learning models, and their security models they use are not necessarily aligned. We now briefly outline the techniques and security guarantees that are claimed in each of the works. Our goal is not to provide an exhaustive survey, but rather to illustrate the landscape through recent works that are broadly representative of the field—even though they span different years, venues, and communities (ranging from machine learning to security).

## D.1 Zero Knowledge Proofs for Decision Tree Predictions and Accuracy

**Goal and Solution Details.** Zhang et al. (2020) introduce protocols for auditing accuracy and verifying decision tree predictions. These protocols enable the owner of a decision tree model to prove that the model produces a given prediction on a data sample, or that it obtains a specified accuracy on a given dataset, without revealing any additional information about the model itself. Zhang et al. (2020)'s main contribution is in designing a custom zero-knowledge proof tailored to efficiently verifying the decision tree prediction. The proof consists of algorithms to generate public parameters, custom commitment algorithm for decision tree models, the prover's algorithm which outputs a proof of inference/accuracy, and verifier algorithm to check this proof. The prover, i.e., model provider, must first commit to its model and subsequently demonstrate that the predictions on client queries are consistent with this commitment. For accuracy verification, the authors propose a batching technique that allows to more efficiently checks the correctness of predictions across *multiple* inputs. They then add an extra verification step to determine how many of these predictions match the true labels.

**Security Model.** Zhang et al. (2020)'s security definition is formulated for the case of inference, and follows the traditional zero-knowledge definition structure, which considers two parties (prover and verifier), and where the protocol is required to satisfy correctness, soundness, and zero-knowledge. Either of the two parties can be malicious. In the context of our analysis we are interested in soundness, which specifies whether a malicious prover can deceive the verifier (i.e., auditor), that the prover's hypothesis passed the test. At a high level, the authors' soundness definition can be summarized as follows: A prover should not be able to output a commitment to a tree $\mathcal{T}$ along with a proof $\pi$, prediction $y$ and datapoint $a$ such that the verifier accepts the proof and at the same time, the $\mathcal{T}'s$ prediction for $a$ is not equal to $y$. Definition of soundness for the accuracy case is similar: the prover outputs the dataset which is used for checking accuracy, and wins the game if the verifier accepts the proof even though the accuracy is not what the prover claims it to be.

**Discussion.** The security notion in this work aligns well with the intuitive goals of verifying both the correctness of individual predictions and the accuracy of a model on a given dataset. However, it does not give any formal guarantees for datasets beyond the audited dataset, i.e., the accuracy verification solution does not generalize to other datasets drawn from the same distribution. In fact, Zhang et al. (2020) explicitly note that it is possible to use their solution to check accuracy on a *public dataset*. In this setting, their approach falls within our framework of Definition 3, and is vulnerable to the same attack as outlined in §4.1. In fact, note that our example works even given an *ideal* proof of accuracy (when it is checked on a dataset known to the adversary), and even if the prover supplies an *additional* proof of training to complement its proof of accuracy.

## D.2 P2NIA: Privacy-Preserving Non-Iterative Auditing

**Goal and Solution Details.** Bourrée et al. (2025) propose a novel auditing scheme that enables one-shot verification of a model's group fairness while preserving privacy for both parties: the model provider is not required to open-source the model, and the auditor need not disclose any private information to support the audit. The main contribution of Bourrée et al. (2025) is a mechanism that enables auditing without requiring the auditor to supply the audit dataset. Specifically, the model provider supplies a dataset together with the corresponding predictions (both in the clear), which the auditor then uses to verify the fairness condition. To construct this dataset, model provider draws on a portion of its internal training data. To preserve confidentiality of this data, it is not shared directly. Instead, model provider feeds it into a synthetic data generation algorithm, and the resulting synthetic dataset is what is sent to the auditor.

**Security Model.** The work does not provide a formal security model. It is set up in the black-box setting and assumes that the auditor does not know the distribution of the model owner's training data.

**Discussion.** As Bourrée et al. (2025) do not utilize cryptographic techniques to prove that the outputs actually correspond to the given inputs, the prover can easily cheat by simply adjusting the labels it supplies for the constructed dataset. However, even if one were to strengthen the scheme by adding a secure proof of training (e.g., Pappas and Papadopoulos (2024)) together with inference proofs (as in Zhang et al. (2020)), the fact that the model owner knows the dataset that is being used for the audit means that the solution falls within our framework of Definition 3, and is thus vulnerable to data-forging attacks. An interesting open question would be to see if, since in this scenario the model owner not only knows, but directly influences the audit dataset, there can be an even simpler attack.

### D.3 CONFIDENTIAL-PROFITT: CONFIDENTIAL PROof of FAIr Training of Trees

**Goal and Solution Details.** Shamsabadi et al. (2022) propose Confidential-PROFITT, a framework for certifying fairness of decision trees while preserving confidentiality of both the model and the training data. Confidential-PROFITT consists of a zero-knowledge-friendly decision tree learning algorithm that, when executed honestly, enforces fairness by design—up to a tunable degree controlled by a parameter. On top of this, Confidential-PROFITT designs a zero-knowledge proof system to verify fairness of a decision tree. The proof requires the model provider to commit to both the model and its training data, then prove in zero-knowledge that the paths taken by the committed training points through the (committed) decision tree satisfy specified fairness bounds. In terms of fairness metrics, Confidential-PROFITT supports *demographic parity* and *equalized odds* as fairness metrics.

**Security Model.** Confidential-PROFITT considers a malicious model provider (that, however, is assumed to commit to the training data honestly) and a malicious auditor (who wishes to learn model details/training data), and obtains standard zero-knowledge proof properties (correctness, soundness, zero-knowledge) with respect to a statement that can be summarized roughly as follows "With respect to a private dataset *chosen by the model provider*, the committed model satisfies certain fairness guarantees".

**Discussion.** Confidential-PROFITT assumes that the model provider honestly commits to the training data. Under this assumption, the corresponding zero-knowledge proof certifies that the resulting model inherits the fairness guarantees of the fair learning algorithm introduced in Confidential-PROFITT (which the authors show indeed improves fairness). However, if the provider is not restricted to committing to the true training data, Confidential-PROFITT is vulnerable to data-forging attacks, as the provider can choose the audit dataset before committing to the model.

### D.4 OATH: EFFICIENT AND FLEXIBLE ZERO-KNOWLEDGE PROOFS OF END-TO-END ML FAIRNESS

**Goal and Solution Details.** Franzese et al. (2024) present OATH, a model-agnostic fairness auditing framework. The core idea in OATH is to leverage clients (who query the model during deployment) to participate in the auditing process. OATH operates in two phases: (i) a certification protocol between the model provider and the auditor, and (ii) a query authentication protocol involving model provider, inference clients, and auditor (dubbed verifier in OATH). The first phase follows the standard certification flow we describe in §3. In the second phase, the auditor receives commitments to client queries and the corresponding model predictions. These commitments can later be verified in zero knowledge for fairness, correctness, and consistency with the certified model.

**Security Model.** OATH considers three fully malicious entities: a model provider, inference clients, and an auditor. These parties are assumed not to collude with each other. The auditor assesses model fairness both with respect to the calibration dataset and the clients queries. The system provides standard correctness, soundness, and zero-knowledge with respect to these two datasets.

**Discussion.** The calibration dataset which is used in the certification protocol between the model provider and the auditor might be supplied by either party. If the calibration dataset is chosen by the prover, same as P2NIA and Confidential-PROFITT, the corresponding fairness check is vulnerable to data forging. However, in contrast to prior works, OATH can fall back on guarantees based on client's queries.

### D.5 FairProof: Confidential and Certifiable Fairness for Neural Networks

**Goal and Solution Details.** Yadav et al. (2024) propose FairProof, a fairness certification approach that maintains confidentiality of the model. In contrast to Confidential-PROFITT and OATH, which focus on group fairness metrics, FairProof considers local individual fairness. This allows Yadav et al. (2024) to issue a personalized certificate to every client.

**Security Model.** FairProof system involves a malicious model provider and malicious clients (who wish to learn model details/training data), and considers standard correctness, soundness, and zero-knowledge properties. The corresponding statement is roughly as follows: "Given a datapoint $x$, the model's output is $y$ and a lower bound on an individual fairness parameter for $x$ is $\epsilon_x$".

**Discussion.** The usage of a specific fairness metric (local individual fairness) allows FairProof to provide per-client certificates of fairness, and escape the problems that arise from the usage of reference datasets (including vulnerability to data-forging attacks). On the flip side, FairProof requires to generate fairness certificates during deployment and does not provide any fairness guarantees prior to deployment.

### D.6 zkCNN: Zero Knowledge Proofs for Convolutional Neural Network Predictions and Accuracy

**Goal and Solution Details.** Liu et al. (2021) propose zkCNN, a zero-knowledge proof protocol for inference and accuracy of convolutional neural networks (CNNs). The core contribution is a novel sumcheck protocol (which is the key ingredient in many zero-knowledge system) that is tailored to two-dimensional convolutions.

**Security Model.** zkCNN considers the standard setting with a prover and a verifier. Either party can be malicious. Liu et al. (2021)'s security definition for inference is a zero-knowledge-style definition, and the scheme is required to satisfy correctness, soundness, and zero-knowledge. Similar to Zhang et al. (2020), Liu et al. (2021)'s soundness intuitively states that a prover should not be able to output a commitment to a model and provide a proof $\pi$, prediction $y$ and datapoint $X$ such that the verifier accepts the proof, and at the same time, the committed model's prediction for $X$ is not equal to $y$. If instantiated with a specific commitment scheme, Liu et al. (2021)'s scheme further satisfies knowledge soundness, the stronger version of soundness where there exists an extractor to extract the CNN parameters from a valid proof and prediction with overwhelming probability. Liu et al. (2021) do not provide a security definition for their proof of accuracy.

**Discussion.** As Liu et al. (2021) do not give a security definition for their proof of accuracy, the formal security guarantee they provide is not fully clear. However, the authors indicate that their scheme can be used to prove the accuracy on a public dataset. This scenario falls within our framework of definition 3, and is vulnerable to the same style of attack as outlined in §4.1.

### D.7 Scaling up Trustless DNN Inference with Zero-Knowledge Proofs

**Goal and Solution Details.** Kang et al. (2022) propose a zero-knowledge-based framework for verifying DNN inference and accuracy. Their key contribution is a careful translation of DNN specifications into arithmetic circuits suitable for zero-knowledge proofs. The system also introduces economic incentives to support ML-as-a-service. Concretely, when verifying accuracy, the model provider first commits to the model, and the client commits to the test set. Both parties then deposit monetary collateral into an escrow. The client reveals the test set, and the provider must produce a zero-knowledge proof that the committed model meets the claimed accuracy. If the provider fails or refuses to prove the required accuracy, it forfeits its collateral; otherwise, the client pays for the service.

**Security Model.** Kang et al. (2022) study the standard two-party setting with a *prover* (model provider) and a *verifier* (client), either of whom may be malicious. Cryptographically, they aim for the standard zero-knowledge proof properties: *completeness*, *knowledge soundness*, and *zero knowledge*. They further consider incentives, showing that—under certain assumptions—honest model providers and clients are motivated to participate in the accuracy verification protocol, while malicious parties are discouraged.

**Discussion.** In terms of cryptographic guarantees, Kang et al. (2022) gets the core design right: their protocol for proofs of accuracy closely follows the framework outlined in §5 and is not vulnerable to our data-forging attacks. However, Kang et al. (2022) provide no formal guarantees about accuracy on data outside the audited set. It would be interesting to perform an analysis similar to that in §E.2.1 given their constraints.

### D.8 EZDPS: AN EFFICIENT AND ZERO-KNOWLEDGE MACHINE LEARNING INFERENCE PIPELINE

**Goal and Solution Details.** Wang and Hoang (2023) introduce ezDPS, a pipeline for zero-knowledge proofs of inference correctness and accuracy above a specified threshold. They construct arithmetic circuit gadgets for key ML operations, including exponentiation, absolute value, and array max/min, and further devise optimized methods for proving Discrete Wavelet Transform, Principal Component Analysis, and multi-class Support Vector Machines with various kernel functions using an efficient set of arithmetic constraints.

**Security Model.** Wang and Hoang (2023) consider two mutually distrusting parties – a malicious server and a semi-honest client, who follows the protocol but aims to learn information about the model's parameters. For their inference pipeline, they consider standard definitions of correctness, soundness, and zero-knowledge (similar to those by Zhang et al. (2020) and Kang et al. (2022)). Wang and Hoang (2023) do not provide a security definition for their proof of accuracy.

**Discussion.** Similar to Liu et al. (2021), as Wang and Hoang (2023) do not provide a security definition for their proof of accuracy, the precise security guarantee they achieve is somewhat unclear. However, Wang and Hoang (2023) indicate that their scheme can be used to prove the accuracy on a public dataset, which falls within our framework of definition 3. This instantiation of their method is vulnerable to the same style of attack as outlined in §4.1.

### D.9 CONFIDENTIAL-DPPROOF: CONFIDENTIAL PROOF OF DIFFERENTIALLY PRIVATE TRAINING

**Goal and Solution Details.** Shamsabadi et al. (2024) present Confidential-DPproof, a framework that enables the model provider to prove to an auditor that their model was correctly trained via DP-SGD, a classic approach for training models with differential privacy guarantees. The certification of DP-SGD's training run is done in zero-knowledge.

**Security Model.** Shamsabadi et al. (2024) consider two mutually distrusting parties: a prover, i.e., model provider, and an auditor. The prover is fully malicious, while the auditor is semi-honest and aims to obtain information about the model's parameters. Confidential-DPproof considers standard definitions of correctness, soundness, and zero-knowledge.

**Discussion.** The data used by Shamsabadi et al. (2024) for their zero-knowledge proof is selected by the prover. This fits the framework in §3, and makes the solution susceptible to data-forging attacks. In particular, a malicious prover could degrade the claimed differential privacy guarantees by, for example, supplying multiple copies of its (otherwise honest) training data as the input to the Confidential-DPproof protocol. We leave a formal treatment and full development of this attack as an interesting direction for future work.

## E SECURE CONSTRUCTION OF AUDITING PROTOCOLS

### E.1 AUDITING PROTOCOL AND SECURITY DEFINITIONS

In this section, we formally define the security properties of an auditing protocol.

**Completeness** An auditing protocol $\Pi$ is complete with error $p$ if for any model $h$ such that $F(h, S_{\text{train}}) = 1$, the following holds:

$$\Pr\left[b = 1 \,:\, (\mathsf{com}, b) \leftarrow \langle \mathsf{Prove}(h, S_{\text{train}}), \mathsf{Audit} \rangle\right] \geq 1 - p$$

**Binding** An auditing protocol $\Pi$ is *computationally binding* if for any PPT adversary $\mathcal{A}$, the following is negligible in $\lambda$:

$$\Pr\left[\begin{array}{c} \mathsf{Commit}(h||S_{\text{train}}; \rho) = \mathsf{Commit}(h'||S'_{\text{train}}; \rho') \\ \wedge\ (h \neq h' \vee\ S_{\text{train}} \neq S'_{\text{train}}) \end{array} : (h, h', S_{\text{train}}, S'_{\text{train}}, \rho, \rho') \leftarrow \mathcal{A}(1^\lambda)\right]$$

Note that we require the binding property to hold both for the model and the training dataset. This can be easily achieved by separately committing to the model and the training dataset with a standard binding commitment scheme (§ A.5), and then outputting the concatenation of the two commitments.

$\tilde{F}$**-Relaxed Knowledge Soundness** An auditing protocol $\Pi$ is $\tilde{F}$-relaxed knowledge sound with knowledge error $\kappa$ if for any PPT adversary $\mathsf{Prove}^*$, there exists an expected polynomial time extractor $\mathsf{Ext}_{\mathsf{Prove}^*}$ such that the following holds:

$$p_{\text{ext}} \geq p_{\text{acc}} - \kappa$$

where

$$p_{\text{ext}} = \Pr\left[\begin{array}{l} (\mathsf{com} = \mathsf{Commit}(h||S_{\text{train}}; \rho) \wedge\ \tilde{F}(h) = 1) \\ \vee\ \big(\mathsf{com} = \mathsf{Commit}(h||S_{\text{train}}; \rho) \\ \quad \wedge\ \mathsf{com} = \mathsf{Commit}(h'||S'_{\text{train}}, \rho') \\ \quad \wedge\ (h \neq h' \vee\ S_{\text{train}} \neq S'_{\text{train}})\big) \end{array} : \begin{array}{l} (\mathsf{com}, b) \leftarrow \langle \mathsf{Prove}^*, \mathsf{Audit} \rangle; \\ (h, S_{\text{train}}, \rho, h', S'_{\text{train}}, \rho') \leftarrow \mathsf{Ext}_{\mathsf{Prove}^*}(\mathsf{com}) \end{array}\right]$$

$$p_{\text{acc}} = \Pr\left[b = 1 : (\mathsf{com}, b) \leftarrow \langle \mathsf{Prove}^*, \mathsf{Audit} \rangle\right]$$

Intuitively, this notion guarantees that if a cheating prover $\mathsf{Prove}^*$ convinces the auditor to accept with non-negligible probability, then it must either know a model $h$ and a training dataset $S_{\text{train}}$ satisfying a predicate $\tilde{F}$, **or** find two distinct openings to the same commitment com. As the latter event happens with negligible probability if Commit is computationally binding, this implies that $\mathsf{Prove}^*$ must know a valid model $h$ and a training dataset $S_{\text{train}}$ satisfying $\tilde{F}$. We call this property "$\tilde{F}$-relaxed" knowledge soundness because the predicate $\tilde{F}$ is a relaxation of the original predicate $F$. This is necessary because the auditor only checks the property $f$ on a *finite sample*, which may not perfectly reflect the property $F$ on the *underlying distribution*. In our concrete instantiations, we will quantify the gap between $F$ and $\tilde{F}$ (see § E.2.1 and § E.2.2).

**Zero Knowledge** Let $\mathsf{view}_{\mathsf{Audit}}^{\mathsf{Prove}(h, S_{\text{train}})}$ be a string consisting of all the incoming messages that Audit receives from Prove during the interaction $\langle \mathsf{Prove}(h, S_{\text{train}}), \mathsf{Audit} \rangle$, and Audit's random coins. $\Pi$ is *zero-knowledge* against semi-honest auditor if there exists a PPT simulator Sim such that for any PPT adversary $\mathcal{A}$, and any $h$ such that $F(h, S_{\text{train}}) = 1$, the following is negligible in $\lambda$.

$$\left| \Pr\left[b = 1 : b \leftarrow \mathcal{A}(\mathsf{view}_{\mathsf{Audit}}^{\mathsf{Prove}(h, S_{\text{train}})})\right] - \Pr\left[b = 1 : \mathsf{view}' \leftarrow \mathsf{Sim}(1^\lambda);\ \ b \leftarrow \mathcal{A}(\mathsf{view}')\right] \right|$$

### E.2 CONSTRUCTION OF AUDITING PROTOCOL

We construct a commit-sample-prove auditing scheme $\Pi_{\text{csp}}$. While we focus on a protocol checking $F$ on a hypothesis $h$ only, the construction below can be naturally extended to a more complex $F$ that additionally takes a training dataset as input. Let Commit be a binding commitment scheme (A.5) and $\mathsf{ZKP} = (\mathcal{P}, \mathcal{V})$ be a ZK proof system for the following relation $\mathcal{R}$: for a pair of public statement $\mathsf{x} = (\mathsf{com}, S_{\text{audit}})$ and private witness $\mathsf{w} = (h, \rho)$, we have $(\mathsf{x}, \mathsf{w}) \in \mathcal{R} \iff f(h, S_{\text{audit}}) = 1 \wedge \mathsf{com} = \mathsf{Commit}(h; \rho)$. We define a commit-sample-prove auditing protocol $\Pi_{\text{csp}} = (\mathsf{Commit}, \mathsf{Prove}, \mathsf{Audit})$ using an emprical predicate $f$ and a distribution $\mathcal{D}$ over a query space $Q = \{(x_i, y_i)\}_{i=1}^m$ as follows:

$\langle \mathsf{Prove}(h), \mathsf{Audit} \rangle$

1. Prove computes $\mathsf{com} = \mathsf{Commit}(h; \rho)$ using a uniformly random string $\rho \in \{0, 1\}^{l_{\mathsf{Commit}}}$ and sends com to Audit.

2. Audit samples $S_{\text{audit}} \leftarrow \mathcal{D}^n$ and sends it to Prove.

3. Prove and Audit execute $b \leftarrow \langle \mathcal{P}(\mathsf{w}), \mathcal{V} \rangle(\mathsf{x})$, where $\mathsf{x} = (\mathsf{com}, S_{\text{audit}})$ and $\mathsf{w} = (h, \rho)$. Here, Prove plays $\mathcal{P}$ and Audit plays $\mathcal{V}$.

4. Prove outputs com, while Audit outputs $b$.

We now state our main theorem regarding the security of $\Pi_{\mathsf{csp}}$. The result is stated for a general auditing task defined by a predicate $F$ and an empirical predicate $f$.

**Theorem 3.** *Suppose the empirical predicate $f$, the model predicate $F$, and the relaxed model predicate $\tilde{F}$ satisfy the following false negative and false positive rate bounds for every model $h$:*

$$\Pr_{S_{audit} \leftarrow \mathcal{D}^n}[f(h, S_{audit}) \neq 1 \mid F(h) = 1] \leq p_{fnr}$$

$$\Pr_{S_{audit} \leftarrow \mathcal{D}^n}[f(h, S_{audit}) = 1 \mid \tilde{F}(h) \neq 1] \leq p_{fpr}$$

*Then $\Pi_{csp}$ is a secure auditing protocol for $F$ satisfying the following properties:*

- *If ZKP is perfectly complete, then $\Pi_{csp}$ is complete with error $p_{fnr}$ for any model $h$ such that $F(h) = 1$.*

- *If the underlying commitment scheme Commit is computationally binding, then $\Pi_{csp}$ is computationally binding.*

- *If ZKP is knowledge sound with knowledge error $\kappa$, then $\Pi_{csp}$ is $\tilde{F}$-relaxed knowledge sound with knowledge error $\kappa + p_{fpr}$.*

- *If $p_{fnr}$ is negligible in $\lambda$, Commit is hiding, and ZKP is zero-knowledge, then $\Pi_{csp}$ is zero-knowledge against semi-honest auditor.*

*Proof sketch.* The proof carefully combines the standard arguments in learning theory and the security properties of the underlying cryptographic primitives. We focus on knowledge soundness, as completeness, binding and zero-knowledge directly follow from the corresponding properties of the underlying commitment scheme and ZK proof system. To argue knowledge soundness, let us assume for the sake of simplicity the commitment scheme is perfectly binding and straightline extractable, i.e., once the prover sends com to the auditor, one can immediately extract a unique model $h$ and a unique randomness $\rho$ such that $\mathsf{com} = \mathsf{Commit}(h; \rho)$. Such a commitment scheme can be constructed in the common reference string (CRS) model using public key encryption. If the committed model $h$ does *not* satisfy $\tilde{F}$, then by the assumption on the false positive rate, the probability that $f(h, S_{\mathsf{audit}}) = 1$ is at most $p_{\mathsf{fpr}}$ over the choice of $S_{\mathsf{audit}}$. Note that we need to relax the predicate from $F$ for completeness to $\tilde{F}$ because the auditor only checks the empirical predicate on a finite sample, which may not perfectly reflect the true predicate. Depending on the deployment scenario, this gap can be made arbitrarily small by increasing the sample size $n$.

The actual proof is significantly more involved, when the commitment scheme is only computationally binding (which is the case for most practical instantiations). In particular, we need to construct a meta-extractor that runs the knowledge extractor $\mathcal{E}$ for ZKP to obtain a candidate model $h$, and rewinds $\mathcal{E}$ with fresh $S'_{\mathsf{audit}} \sim \mathcal{D}^n$ to ensure the validity of $h$ via probabilistic tests (or otherwise break binding of the commitment). Our formal security proof takes care of these subtle technicalities by leveraging the proof techniques from lattice-based zero knowledge proofs.

*Proof.* Binding trivially follows from the computational binding property of the underlying commitment scheme.

Completeness: By the assumption on $f$ and $F$, an honest auditing prover Prove fails to convince the auditor playing verifier $\mathcal{V}$ after receiving fresh $S_{\mathsf{audit}}$ with probability at most $p_{\mathsf{fnr}}$. Since Commit and ZKP are perfectly complete, an honest ZKP prover $\mathcal{P}$ given any valid witness always convinces the verifier. Thus, the overall completeness error is at most $p_{\mathsf{fnr}}$.

Knowledge Soundness: The protocol can be viewed as a commit-and-prove zero-knowledge proof with interleaved probabilistic tests on the statement. This approach is common in lattice-based zero-knowledge proofs (Cf. Theorem 5.1.6 of Nguyen (2022)).

Let Prove* be any PPT adversary. We denote by $\mathcal{P}_1$ the interactive ZK prover algorithm that Prove* invokes in Step 3 to prove the statement x fixed by the previous steps. Moreover, denote by $R_{\mathcal{E}}$

(resp. $R_{\mathcal{P}}$) the randomness space of the extractor $\mathcal{E}$ (resp. prover $\mathcal{P}_1$) for ZKP. We first construct the following extractor:

$\underline{\mathsf{Ext}_{\mathsf{Prove}^*}}$

1. Run $\mathsf{Prove}^*$ to get com.

2. Sample $S_{\mathrm{audit}} \leftarrow \mathcal{D}^n$, $r_{\mathcal{E}} \leftarrow R_{\mathcal{E}}$, and $r_{\mathcal{P}} \leftarrow R_{\mathcal{P}}$.

3. Let $\mathcal{P}_0$ be the algorithm that outputs $\mathsf{x} = (\mathsf{com}, S_{\mathrm{audit}})$ as a statement and $\mathcal{P} = (\mathcal{P}_0, \mathcal{P}_1)$, where $\mathcal{P}_1$'s randomness is fixed to $r_{\mathcal{P}}$. Run $\mathcal{E}_{\mathcal{P}}(\mathsf{x}; r_{\mathcal{E}})$ to extract the witness $(h, \rho)$.

4. If $f(h, S_{\mathrm{audit}}) \neq 1$ or $\mathsf{com} \neq \mathsf{Commit}(h; \rho)$, abort.

5. Repeat the following process:

    (a) Sample $S'_{\mathrm{audit}} \leftarrow \mathcal{D}^n$ and $r'_{\mathcal{E}} \leftarrow R_{\mathcal{E}}$.
    (b) Let $\mathcal{P}'_0$ be the algorithm that outputs $\mathsf{x}' = (\mathsf{com}, S'_{\mathrm{audit}})$ as a statement and $\mathcal{P}' = (\mathcal{P}'_0, \mathcal{P}_1)$, where $\mathcal{P}_1$'s randomness is fixed to $r_{\mathcal{P}}$. Run $\mathcal{E}_{\mathcal{P}'}(\mathsf{x}'; r'_{\mathcal{E}})$ to extract the witness $(h', \rho')$.
    (c) If $f(h', S'_{\mathrm{audit}}) = 1$ and $\mathsf{com} = \mathsf{Commit}(h'; \rho')$, terminate and output $(h, \rho, h', \rho')$
    (d) Else, go to step (a).

**Running time:**

Let $T$ be the random variable counting the number of calls to the inner extractor $\mathcal{E}$ until termination. For each fixed com and prover's randomness $i \in R_{\mathcal{P}}$ we denote by $\epsilon_i$ the probability that $\mathcal{E}_{\mathcal{P}}((\mathsf{com}, S); r_{\mathcal{E}})$ successfully outputs $(h, \rho)$ with $f(h, S) = 1$ and $\mathsf{com} = \mathsf{Commit}(h; \rho)$, where the probability is taken over $r_{\mathcal{E}} \leftarrow R_{\mathcal{E}}$ and $S \sim \mathcal{D}^n$. Denote by $E$ the event that $\mathcal{E}_{\mathcal{P}}$ successfully outputs a valid witness at Step 3. We now evaluate the expected running time of Ext as follows:

$$\mathbb{E}[T] = \sum_{i \in R_{\mathcal{P}}} \left( \mathbb{E}[T \mid r_{\mathcal{P}} = i \wedge E] \cdot \Pr[r_{\mathcal{P}} = i \wedge E] + \mathbb{E}[T \mid r_{\mathcal{P}} = i \wedge \neg E] \cdot \Pr[r_{\mathcal{P}} = i \wedge \neg E] \right)$$

$$= \frac{1}{|R_{\mathcal{P}}|} \cdot \sum_{i \in R_{\mathcal{P}}} \left( \mathbb{E}[T \mid r_{\mathcal{P}} = i \wedge E] \cdot \Pr[E \mid r_{\mathcal{P}} = i] + \mathbb{E}[T \mid r_{\mathcal{P}} = i \wedge \neg E] \cdot \Pr[\neg E \mid r_{\mathcal{P}} = i] \right)$$

$$= \frac{1}{|R_{\mathcal{P}}|} \cdot \sum_{i \in R_{\mathcal{P}}} \left( \mathbb{E}[T \mid r_{\mathcal{P}} = i \wedge E] \cdot \epsilon_i + \mathbb{E}[T \mid r_{\mathcal{P}} = i \wedge \neg E] \cdot (1 - \epsilon_i) \right)$$

$$\leq \frac{1}{|R_{\mathcal{P}}|} \cdot \left( \sum_{i \in R_{\mathcal{P}}} \left( \frac{1}{\epsilon_i} + 1 \right) \cdot \epsilon_i + 1 \cdot (1 - \epsilon_i) \right) = 2$$

where we used the fact that $\mathbb{E}[T \mid r_{\mathcal{P}} = i \wedge \neg E] = 1$ since the algorithm terminates after the initial extraction fails. Moreover, since the underlying ZKP is knowledge sound, each call to $\mathcal{E}$ runs in expected polynomial time. Overall, we conclude that Ext runs in expected polynomial time.

**Knowledge error:** We define the following events:

- $E$: $\mathsf{com} = \mathsf{Commit}(h; \rho) \wedge f(h, S_{\mathrm{audit}}) = 1$

- $E'$: $\mathsf{com} = \mathsf{Commit}(h'; \rho') \wedge f(h', S'_{\mathrm{audit}}) = 1$

- $E_1$: $E \wedge E'$

- $E_2$: $h = h'$

- $E_3$: $\tilde{F}(h) = 1$

Our goal is to relate the success probability $p_{\text{ext}}$ of the extractor to $p_{\text{acc}}$, where

$$p_{\text{ext}} = \Pr[E_1 \wedge (\neg E_2 \vee E_3)]$$
$$p_{\text{acc}} = \Pr[b = 1 \,:\, (\mathsf{com}, b) \leftarrow \langle \mathsf{Prove}^*, \mathsf{Audit} \rangle]$$

To this end, we first rewrite $p_{\text{ext}}$ as follows:

$$p_{\text{ext}} = \Pr[E_1 \wedge (\neg E_2 \vee E_3)] = \Pr[E_1] - \Pr[E_1 \wedge E_2 \wedge \neg E_3]$$

We now bound $\Pr[E_1]$ and $\Pr[E_1 \wedge E_2 \wedge \neg E_3]$ separately.

**Bounding $\Pr[E_1]$:** We rewrite $\Pr[E_1]$ as follows:

$$\Pr[E_1] = \Pr[E] - \Pr[E \wedge \neg E'] = \Pr[E] \geq p_{\text{acc}} - \kappa$$

where $\Pr[E \wedge \neg E'] = 0$ since if $E$ happens, then the extractor always terminates in expected polynomial time and outputs $(h, \rho, h', \rho')$ at Step 5, which implies that $E'$ also happens. The last inequality follows from the definition of $p_{\text{acc}}$ and the knowledge soundness of ZKP.

**Bounding $\Pr[E_1 \wedge E_2 \wedge \neg E_3]$:** For each fixed com and $i \in R_{\mathcal{P}}$, let $\epsilon_i$ be the probability that fresh $(r_{\mathcal{E}}, S) \leftarrow R_{\mathcal{E}} \times \mathcal{D}^n$ leads to successful extraction.

We first rewrite $\Pr[E_1 \wedge E_2 \wedge \neg E_3]$ as follows:

$$\Pr[E_1 \wedge E_2 \wedge \neg E_3] = \sum_{i \in R_{\mathcal{P}}} \Pr[r_{\mathcal{P}} = i] \cdot \Pr[E \wedge E' \wedge E_2 \wedge \neg E_3 | r_{\mathcal{P}} = i]$$

$$= \frac{1}{|R_{\mathcal{P}}|} \sum_{i \in R_{\mathcal{P}}} \Pr[E \wedge E' | r_{\mathcal{P}} = i] \cdot \Pr[E_2 \wedge \neg E_3 \,|\, r_{\mathcal{P}} = i \wedge E \wedge E']$$

$$\leq \frac{1}{|R_{\mathcal{P}}|} \sum_{i \in R_{\mathcal{P}}} \epsilon_i \cdot \Pr[E_2 \wedge \neg E_3 \,|\, r_{\mathcal{P}} = i \wedge E \wedge E']$$

$$= \frac{1}{|R_{\mathcal{P}}|} \sum_{i \in R_{\mathcal{P}}} \epsilon_i \cdot \frac{\Pr[E' \wedge E_2 \wedge \neg E_3 \,|\, r_{\mathcal{P}} = i \wedge E]}{\Pr[E' \,|\, r_{\mathcal{P}} = i \wedge E]}$$

$$= \frac{1}{|R_{\mathcal{P}}|} \sum_{i \in R_{\mathcal{P}}} \epsilon_i \cdot \frac{\Pr[E' \wedge E_2 \wedge \neg E_3 \,|\, r_{\mathcal{P}} = i \wedge E]}{\epsilon_i}$$

$$= \frac{1}{|R_{\mathcal{P}}||R_{\mathcal{E}}|} \sum_{i \in R_{\mathcal{P}}, j \in R_{\mathcal{E}}} \Pr[E' \wedge E_2 \wedge \neg E_3 \,|\, r_{\mathcal{P}} = i \wedge r_{\mathcal{E}} = j \wedge E]$$

$$\leq \frac{1}{|R_{\mathcal{P}}||R_{\mathcal{E}}|} \sum_{i \in R_{\mathcal{P}}, j \in R_{\mathcal{E}}} \Pr[f(h', S'_{\text{audit}}) = 1 \wedge h = h' \wedge \tilde{F}(h) \neq 1 \,|\, r_{\mathcal{P}} = i \wedge r_{\mathcal{E}} = j \wedge E]$$

$$\leq \frac{1}{|R_{\mathcal{P}}||R_{\mathcal{E}}|} \sum_{i \in R_{\mathcal{P}}, j \in R_{\mathcal{E}}} \Pr[f(h, S'_{\text{audit}}) = 1 \,|\, r_{\mathcal{P}} = i \wedge r_{\mathcal{E}} = j \wedge E \wedge \tilde{F}(h) \neq 1]$$

$$\leq p_{\text{fpr}}$$

where the last inequality follows from the assumption on $f$ and $\tilde{F}$, as at this stage $h$ is fixed by event $E$ and elements in $S'_{\text{audit}}$ are sampled independently from $\mathcal{D}^n$.

Combining the bounds, we get

$$p_{\text{ext}} = \Pr[E_1] - \Pr[E_1 \wedge E_2 \wedge \neg E_3]$$
$$\geq p_{\text{acc}} - \kappa - p_{\text{fpr}}$$

This completes the proof.

Zero Knowledge: We construct the following simulator, internally using the simulator $\mathcal{S}$ for ZKP:

$\mathsf{Sim}(1^\lambda)$

1. Generate a dummy commitment $\mathsf{com} \leftarrow \mathsf{Commit}(0; \rho)$ using a uniformly random string $\rho \in \{0,1\}^{\ell_\rho}$.

2. Sample $S_{\text{audit}} \leftarrow \mathcal{D}^n$.

3. Run $\mathcal{S}((\text{com}, S_{\text{audit}}))$ to get a simulated view$'$ for ZKP.

4. Output view $= (\text{com}, S_{\text{audit}}, \text{view}')$.

Since Commit is hiding, the dummy commitment is indistinguishable from a real commitment. Moreover, the output of $\mathcal{S}$ is indistinguishable from the view of Audit during the interaction $\langle \mathcal{P}(h), \mathcal{V} \rangle((\text{com}, S_{\text{audit}}))$ for any valid witness $h$. Since a real execution of $\Pi_{\text{csp}}$ defines $x = (\text{com}, S_{\text{audit}})$ and $w = h$ such that $\mathcal{R}(x, w) = 1$ except with probability at most $p_{\text{fnr}}$, by setting $p_{\text{fnr}}$ to be negligible in $\lambda$, the output of Sim is indistinguishable from the view of Audit during the interaction $\langle \text{Prove}(h), \text{Audit} \rangle$. This completes the proof. $\qquad \square$

### E.2.1 EXAMPLE INSTANTIATION: $\Pi_{\text{CSP}}$ FOR ACCURACY AUDITING

To instantiate $\Pi_{\text{csp}}$ for accuracy auditing, we consider the empirical and true accuracy as follows:

$$\hat{\ell}_S(h) = \frac{1}{n} \sum_{(x,y) \in S} \mathbb{I}(h(x) \neq y)$$

$$\ell(h) = \mathbb{E}_{(x,y) \sim \mathcal{D}}[\mathbb{I}(h(x) \neq y)]$$

where $n = |S|$, and define the empirical predicate $f$, the model predicate $F$, and the relaxed model predicate $\tilde{F}$ as follows.

$$f(h, S_{\text{audit}}) = 1 \iff \hat{\ell}_S(h) \leq t + \delta$$
$$F(h) = 1 \iff \ell(h) \leq t$$
$$\tilde{F}(h) = 1 \iff \ell(h) \leq t + 2\delta$$

To apply Theorem 3 to accuracy auditing, it would be sufficient to find the false negative rate $p_{\text{fnr}}$ and false positive rate $p_{\text{fpr}}$ by the following lemma, and then set $n\delta^2 \in \Omega(\lambda)$.

**Lemma 3.** *For any hypothesis $h$,*

$$\Pr_{S_{\text{audit}} \leftarrow \mathcal{D}^n}[f(h, S_{\text{audit}}) \neq 1 \mid F(h) = 1] \leq 2e^{-2n\delta^2} \tag{1}$$

$$\Pr_{S_{\text{audit}} \leftarrow \mathcal{D}^n}[f(h, S_{\text{audit}}) = 1 \mid \tilde{F}(h) \neq 1] \leq 2e^{-2n\delta^2} \tag{2}$$

*Proof.* Consider the empirical error $\hat{\ell}_S(h) = \frac{1}{n} \sum_{(x,y) \in S} \mathbb{I}(h(x) \neq y)$ and the true error $\ell(h) = \mathbb{E}_{(x,y) \sim \mathcal{D}}[\mathbb{I}(h(x) \neq y)]$. Since each element of $S_{\text{audit}}$ is sampled i.i.d. from $\mathcal{D}$, by Hoeffding's inequality, we have

$$\Pr_{S_{\text{audit}} \leftarrow \mathcal{D}^n}[|\hat{\ell}_S(h) - \ell(h)| \geq \delta] \leq 2e^{-2n\delta^2}.$$

Thus, for any $h$ such that $F(h) = 1$ (i.e., $\ell(h) \leq t$), the probability that $f(h, S_{\text{audit}}) \neq 1$ (i.e., $\hat{\ell}_S(h) > t + \delta$) is at most $2e^{-2n\delta^2}$.

Similarly, for any $h$ such that $\tilde{F}(h) \neq 1$ (i.e., $\ell(h) > t + 2\delta$), the probability that $f(h, S_{\text{audit}}) = 1$ (i.e., $\hat{\ell}_S(h) \leq t + \delta$) is at most $2e^{-2n\delta^2}$. $\qquad \square$

Overall, by Theorem 3 and Lemma 3, we conclude that $\Pi_{\text{csp}}$ instantiated for accuracy auditing is a secure auditing protocol for accuracy with the following properties for a sufficiently large $n\delta^2 \in \Omega(\lambda)$:

- If the true error of the model $h$ is $\leq t$, then the auditor accepts with high probability.
- If the auditor accepts, then the auditor gets assurance that the true error of the model $h$ is at most $t + 2\delta$, even if the prover is misbehaving.

### E.2.2 EXAMPLE INSTANTIATION: $\Pi_{\mathrm{CSP}}$ FOR DEMOGRAPHIC PARITY AUDITING

Similarly, we can instantiate the protocol $\Pi_{\mathrm{csp}}$ for auditing fairness conditions. Take the demographic parity as an example, for which we can set the query space to $Q = \{x_i\}_{i=1}^m$ without ground-truth labels. We consider the empirical and true demographic parity differences as follows:

$$\Delta_{\mathsf{dp}}(h, S_{\mathrm{audit}}) = \left| \frac{1}{n_0} \sum_{x \in S_0} \mathbb{I}(h(x) = 1) - \frac{1}{n_1} \sum_{x \in S_1} \mathbb{I}(h(x) = 1) \right|$$

$$\hat{\Delta}_{\mathsf{dp}}(h) = |\mathrm{E}_{x \sim \mathcal{D}}[\mathbb{I}(h(x) = 1) \,|\, s_x = 0] - \mathrm{E}_{x \sim \mathcal{D}}[\mathbb{I}(h(x) = 1) \,|\, s_x = 1]|$$

where $s_x$ denotes the sensitive feature of a data point $x$, $S_0 = \{x \in S_{\mathrm{audit}} : s_x = 0\}$, $S_1 = \{x \in S_{\mathrm{audit}} : s_x = 1\}$, $n_0 = |S_0|$, and $n_1 = |S_1|$.

Define the empirical predicate $f$, the model predicate $F$, and the relaxed model predicate $\tilde{F}$ as follows.

$$f(h, S_{\mathrm{audit}}) = 1 \iff \Delta_{\mathsf{dp}}(h, S_{\mathrm{audit}}) \leq t + 2\delta$$

$$F(h) = 1 \iff \hat{\Delta}_{\mathsf{dp}}(h) \leq t$$

$$\tilde{F}(h) = 1 \iff \hat{\Delta}_{\mathsf{dp}}(h) \leq t + 4\delta$$

To apply Theorem 3 to demographic parity auditing, we can prove the following lemma in place of Lemma 3, and then set $n_{\min}\delta^2 \in \Omega(\lambda)$.

**Lemma 4.** *For any hypothesis $h$,*

$$\Pr_{S_{\mathrm{audit}} \leftarrow \mathcal{D}^n}[f(h, S_{\mathrm{audit}}) \neq 1 \mid F(h) = 1] \leq 4e^{-2n_{\min}\delta^2} \tag{3}$$

$$\Pr_{S_{\mathrm{audit}} \leftarrow \mathcal{D}^n}[f(h, S_{\mathrm{audit}}) = 1 \mid \tilde{F}(h) \neq 1] \leq 4e^{-2n_{\min}\delta^2} \tag{4}$$

*where $n_{\min} = \min(n_0, n_1)$.*

*Proof.* We first prove (4). Define the following variables:

$$g_0 = \frac{1}{n_0} \sum_{x \in S_0} \mathbb{I}(h(x) = 1) \qquad\qquad p_0 = \mathrm{E}_{x \sim \mathcal{D}}[\mathbb{I}(h(x) = 1) \,|\, s_x = 0]$$

$$g_1 = \frac{1}{n_1} \sum_{x \in S_1} \mathbb{I}(h(x) = 1) \qquad\qquad p_1 = \mathrm{E}_{x \sim \mathcal{D}}[\mathbb{I}(h(x) = 1) \,|\, s_x = 1]$$

By Hoeffding's inequality, we have

$$\Pr[|g_0 - p_0| \geq \delta] \leq 2e^{-2n_0\delta^2}$$

$$\Pr[|g_1 - p_1| \geq \delta] \leq 2e^{-2n_1\delta^2}$$

where the probability is taken over the randomness of $S_{\mathrm{audit}} \leftarrow \mathcal{D}^n$. Thus, for any $h$ such that $\tilde{F}(h) \neq 1$ (i.e., $|p_0 - p_1| > t + 4\delta$), the probability that $f(h, S_{\mathrm{audit}}) = 1$ (i.e., $|g_0 - g_1| \leq t + 2\delta$) can be bounded as follows:

$$\begin{aligned}
\Pr[|g_0 - g_1| \leq t + 2\delta] &\leq \Pr[|g_0 - g_1| \leq |p_0 - p_1| - 2\delta] \\
&\leq \Pr[2\delta \leq |p_0 - p_1 - (g_0 - g_1)|] \\
&\leq \Pr[2\delta \leq |p_0 - g_0| + |p_1 - g_1|] \\
&\leq \Pr[\delta \leq |p_0 - g_0| \vee \delta \leq |p_1 - g_1|] \\
&\leq \Pr[\delta \leq |p_0 - g_0|] + \Pr[\delta \leq |p_1 - g_1|] \\
&\leq 2e^{-2n_0\delta^2} + 2e^{-2n_1\delta^2} \leq 4e^{-2n_{\min}\delta^2}
\end{aligned}$$

Analogously, we can prove (3). For any $h$ such that $F(h) = 1$ (i.e., $|p_0 - p_1| \leq t$), the probability that $f(h, S_{\mathrm{audit}}) \neq 1$ (i.e., $|g_0 - g_1| > t + 2\delta$) can be bounded in the same way:

$$\Pr[|g_0 - g_1| > t + 2\delta] \leq \Pr[|g_0 - g_1| > |p_0 - p_1| + 2\delta] \leq 4e^{-2n_{\min}\delta^2}$$

$\square$

Overall, by Theorem 3 and Lemma 4, we conclude that $\Pi_{\mathrm{csp}}$ instantiated for demographic parity auditing is a secure auditing protocol for demographic parity with the following properties for a sufficiently large $n_{\min}\delta^2 \in \Omega(\lambda)$:

- If the true demographic parity of the model $h$ is $\leq t$, then the auditor accepts with high probability.

- If the auditor accepts, then the auditor gets assurance that the true demographic parity of the model $h$ is at most $t + 4\delta$, even if the prover is misbehaving.

## F  RELATED WORK (CONTINUED)

A number of recent works aim to prove desirable model properties. In terms of *what* these works prove, they can be roughly categorized into proofs of training, inference, accuracy, and fairness. In terms of *how* the corresponding protocols work, recent works on certifiable ML can be categorized as follows:

**Cryptographic approaches** A prolific line of research adapts various cryptographic techniques to certify properties such as accuracy, fairness, etc., without revealing the model's details. The most common technique is *zero-knowledge proofs* (zk proofs), which allow to formally prove that a model satisfies certain properties without revealing anything else about the model. They have been used to certify fairness (Shamsabadi et al., 2022; Yadav et al., 2024; Franzese et al., 2024; Zhang et al., 2025b), inference (Zhang et al., 2020), accuracy (Zhang et al., 2020), and to prove that the model has been trained using a certain algorithm (Abbaszadeh et al., 2024; Garg et al., 2023; Sun et al., 2024; Pappas and Papadopoulos, 2024) (without revealing the training data). Other works (Duddu et al., 2024; Chang et al., 2023) use *secure multi-party computation* (MPC), which allows mutually distrusting parties to jointly compute on private inputs without revealing anything about the inputs apart from the outcome.

**Black box auditing/Statistical testing** These approaches probe a model by submitting inputs, collecting outputs, and analyzing them for undesirable behavior. Tramer et al. (2017); Saleiro et al. (2018) use black-box testing to check for potential unfairness or bias, while (Tan et al., 2018) distill a new model to gain insight into the black box one.

**Outside-the-box auditing** Here the model owner provides access to information beyond query responses, such as source code, documentation (Mitchell et al., 2019), hyperparameters, training data, deployment details, or internal evaluation results.

Finally, we note that our work is related to, but distinct from, data poisoning attacks. We discuss the relationship between the two works below.

## G  CRYPTOGRAPHIC AUDITING OF ML: BACKGROUND AND SUBTLETIES

We outline different categories of proofs that are used in the context of auditing machine learning algorithms. For simplicity, from now on we assume that the *training algorithm is public* (note that making it private only makes the adversary in our attacks stronger, i.e., it could potentially be *easier* for the model owner to perform a data-forging or any other type of attack).

**Proof of Training** A *proof of training* can be viewed as a zero knowledge proof for the following relation $\mathcal{R}$: given $\mathsf{x} = (\mathsf{com}_h, \mathsf{com}_S)$, and $\mathsf{w} = (h, S_{\mathit{train}}, \rho, \rho_h, \rho_S)$, $\mathcal{R}$ outputs 1 if and only if $\mathsf{Train}(S_{\mathit{train}}; \rho) = h$, $\mathsf{com}_h = \mathsf{Commit}(h; \rho_h)$ and $\mathsf{com}_S = \mathsf{Commit}(S_{\mathit{train}}; \rho_S)$, where $\rho$ is the randomness used for training. Here, $\mathsf{Commit}$ is a commitment scheme (§A.5). Intuitively, here the commitment lets the prover fix $h$ and $S_{\mathit{train}}$ up front without revealing them.

**Proof of Inference** A *proof of inference* can be viewed as a special case of zero knowledge proof for the following relation $\mathcal{R}$: given $\mathsf{x} = (\mathsf{com}, x, y)$, and $\mathsf{w} = (h, \rho_h)$, $\mathcal{R}$ outputs 1 if and only if $h(x) = y$ and $\mathsf{com} = \mathsf{Commit}(h; \rho_h)$.

**Auditing using Zero Knowledge Proofs** The strongest form of ZK-based auditing arises when the prover first produces a *proof of training*, thereby showing that a specific committed model instance came from an honest training procedure on a private dataset, and subsequently provides a *proof*

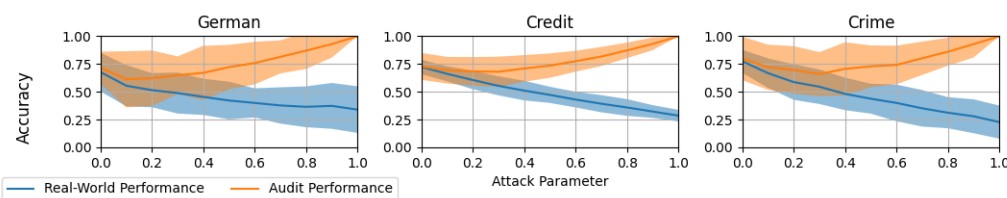

Figure 4: Accuracy of models trained on datasets constructed by Algorithm 1 on various benchmarks. Values are averages over ten runs, error bars represent one standard deviation.

*of property* attesting that the committed model meets the desired criterion. Let $f$ be an auditing function outputting a binary that takes as input a training data set $S_{train}$, an auditing data set $S_{audit}$, and the model $h$. Then privacy-preserving auditing can be realized using zero knowledge proofs for the following relation $\mathcal{R}$: given, $\mathsf{x} = (\mathsf{com}_h, \mathsf{com}_S, S_{audit})$, and $\mathsf{w} = (h, S_{train}, \rho, \rho_h, \rho_S)$, $\mathcal{R}$ outputs 1 if and only if $\mathsf{Train}(S_{train}; \rho) = h$, $f(S_{train}, S_{audit}, h) = 1$, $\mathsf{com}_h = \mathsf{Commit}(h; \rho_h)$ and $\mathsf{com}_S = \mathsf{Commit}(S_{train}; \rho_S)$.

**Definition Subtleties** The zero knowledge property ensures confidentiality of the committed model and training data. However, as we shall see next, knowledge soundness does *not* necessarily capture the actual goal of the auditing process. The reason is that knowledge soundness is typically defined with respect to statements $\mathsf{x} = (\mathsf{com}_h, \mathsf{com}_S, S_{audit})$, which **(1)** are bound to a specific dataset $S_{audit}$, and **(2)** do not specify how or when each component of $\mathsf{x}$ is generated. In practice, it is plausible that $S_{audit}$ is supplied by verifier (i.e., the auditor). We show that if a cheating prover (i.e., model owner) adaptively generates $\mathsf{com}_{h^*}$ and $\mathsf{com}_{S^*}$ *after* observing $S_{audit}$, it is possible to pass the zero knowledge auditing process after maliciously crafting model $h^*$ and/or training data $S^*$. Furthermore, we show that $h^*$ behaves pathologically when evaluated on data outside $S_{audit}$, in a way that completely undermines the purpose of the auditing process.

We note that while this subtlety was indeed overlooked in several works on zero-knowledge-based auditing, it applies even more directly to various non-cryptographic auditing approaches that do not enforce a secure commitment from the prover.

# H   FURTHER EVALUATION

First, we present in Figure 4 results for attacking accuracy audits on additional datasets mentioned in §7.

Next, we present the application of the attack described in § 4.1. In this attack, the adversary is attempting to maximize the model's denial rate $\mathrm{Pr}_{x \sim \mathcal{D}}[h(x) = 0]$ while still appearing accurate to the audit. The results of this attack are given in Figure 5. Observe that as the attack parameter approaches 1 (and the attack becomes maximally malicious), the denial rate of the model on the audit set remains close to the fully honest denial rate while the denial rate on independently sampled data approaches 1. Similarly, the accuracy of the model on the audit set approaches 1, while the true accuracy decreases down to roughly 0.6 (this reflects the true denial rate of the distribution).

Next we examine our ability to observe these attacks by applying statistical tests to the datasets, as described in § C. There is no singular way to determine whether two samples were drawn from the same distribution, so we apply some common statistical tools. In particular, our goal is to determine if the distribution from which the audit data is drawn is identical to the distribution from which the training data is drawn. We use Welch's $t$-test, which serves to determine whether two distributions have the same mean, and Levene's test, a one-way ANOVA for determining whether two distributions have the same variance. These tests are typically applied to 1-dimensional data, and so we apply them to each feature individually. The results of these experiments are given in Table 2.

We observe that the summary statistics of the malicious training data closely match the values for the honest data, suggesting that comparing these two values would not be a successful detection mechanism. This is compounded by the fact that the test statistics for Welch's $t$-test and Levene's test for the malicious training data are considerably smaller on average than the same test statistics for

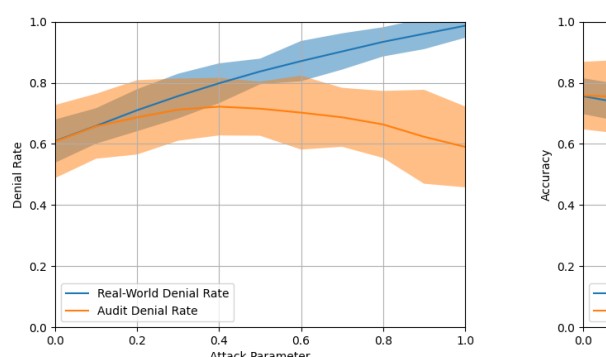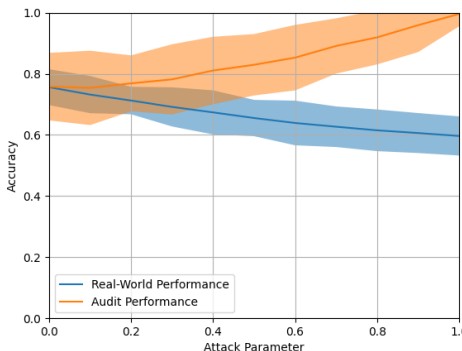

Figure 5: Accuracy and denial rates of models trained on datasets constructed by Algorithm 1 on ACSEmployment. Values are averages over ten runs, error bars represent one standard deviation.

Table 2: Summary and Test statistics for Age feature on ACSEmployment, conditioned on label. Test statistics used are Welch's $t$-test and Levene's test. Attack is undetectable when summary statistics are similar to honest ones, and when test statistics are close to 0. Comparisons are between fully honest and fully malicious datasets.

| Age | | Label = 0 | | Label = 1 | |
|---|---|---|---|---|---|
| | | Honest | Attack | Honest | Attack |
| Summary | $\mu$ | 41.6651 | 41.9657 | 43.9184 | 43.8131 |
| Statistics | $\sigma^2$ | 804.5804 | 810.8822 | 223.1269 | 221.42394 |
| Test | $t$-test | 0.6521 | 0.0033 | 0.7067 | 0.0110 |
| Statistics | ANOVA | 0.6200 | 0.0026 | 1.6500 | 0.0186 |

| Education | | Label = 0 | | Label = 1 | |
|---|---|---|---|---|---|
| | | Honest | Attack | Honest | Attack |
| Summary | $\mu$ | 13.39761692 | 13.41700338 | 18.45539675 | 18.50545506 |
| Statistics | $\sigma^2$ | 42.99789908 | 42.16899485 | 9.979327135 | 8.943082831 |
| Test | $t$-test | 0.7984001575 | 0.0356390553 | 0.9499974697 | 0.1302788154 |
| Statistics | ANOVA | 0.4844657261 | 0.0003374130653 | 1.227829625 | 0.02531893152 |

| Military Status | | Label = 0 | | Label = 1 | |
|---|---|---|---|---|---|
| | | Honest | Attack | Honest | Attack |
| Summary | $\mu$ | 2.5794 | 2.5834 | 3.8121 | 3.8302 |
| Statistics | $\sigma^2$ | 3.2749 | 3.2648 | 0.3507 | 0.3265 |
| Test | $t$-test | 0.4997 | 0.0313 | 0.8699 | 0.1755 |
| Statistics | ANOVA | 1.0240 | 0.0009 | 1.2394 | 0.0304 |

the honest training data, corroborating higher rate of passing the hypothesis tests we observe. At a significance level of $\alpha = 0.05$, we expect a false positive rate of approximately $5\%$. On the other hand, we observe a $0\%$ true positive rate. We note that in a practical application of this attack, the auditor would have access only to the honest or malicious values over a single training run, and would thus be unable to easily distinguish between the two cases by comparing the values or by looking at averages over many runs as we have done here. That being said, an auditor may find it suspicious if the p-value returned by a statistical test is extremely low (even though such a scenario may be very plausible for some distributions); an attacker can safely relax this attack to a comfortable degree, though doing so will increase the risk of failing the audit.

Finally, we present an evaluation of a modified version of the attack that targets neural networks rather than decision trees. Whereas decision trees have very specific conditions that allow us to constrain their behavior, it is much harder to provide theoretical guarantees for neural networks. In order to encourage memorization of the training data, we used a relatively shallow network with very large individual layers. Our attack samples a large amount of training data, and decides whether to

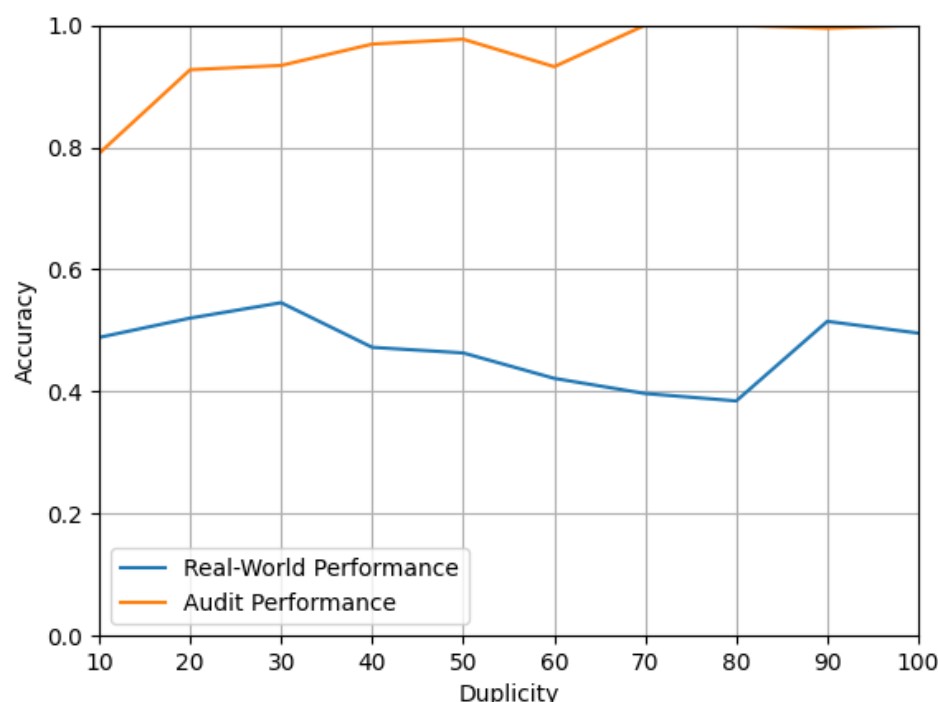

Figure 6: Performance of 226M-parameter neural networks trained on datasets constructed from a mixture of Gaussian distributions. Duplicity refers to the number of perturbed copies of the audit dataset included in the training data.

label each point with the honest label or dishonest label depending on its proximity to the nearest audit data point. We evaluated this attack on an 8-dimensional mixture of Gaussian distributions; the results are shown in Figure 6.

