# OpenReview forum: "Data Forging Attacks on Cryptographic Model Certification"
_ICLR.cc/2026/Conference — Submitted to ICLR 2026_

### Official Review · Reviewer_DHwq · 2025-10-25

**Soundness:** 3
**Presentation:** 3
**Contribution:** 4
**Rating:** 8
**Confidence:** 2

**Summary:**

This paper proposes a novel attack towards cryptographically certified machine learning (ML) models. It shows that certifying ML models with pre-revealed audit datasets can be exploited by an adversary to forge models that satisfy the certification requirements while causing severe mispredictions on fresh inputs. The attack leverages the knowledge of the audit dataset to strategically construct the training data, enabling the adversary to manipulate the model's behavior on unseen data while still passing the certification checks. Experimental results demonstrate that this method can cause ML models to have less than 30% accuracy on fresh inputs while still being certified to have 99% accuracy on the audit dataset.

**Strengths:**

- Novel attack vectors on cryptographically certified ML models, highlighting the severity of data-dependant vulnerabilities.
- Promising methodology to achieve model forgery while maintaining stealth.
- Clear presentation and easy-to-follow methodology for the proposed attack.

**Weaknesses:**

- The threat model can be better explained.
- The evaluation can be extended to include more diverse datasets and model architectures.
- The impact of training dataset size can be further explored.

**Questions:**

This paper proposes an interesting lens into the data-dependant vulnerabilities of certified cryptographic ML models. It brings a new perspective to the security of privacy-preserving ML models. There are, however, several concerns and questions about the proposed attack and its implications:

- **Threat Model Clarification**: The threat model could be better defined. Specifically, what are the capabilities and limitations of the adversary? Why such capabilities are realistic? In particular, if the training dataset is pre-determined and fixed, then this attack would not be feasible. By producing a hashing for all the training data, the adversary cannot freely choose the training data to manipulate the model. That would be an easy defense against this attack.
- **Generalizability of the Attack**: The evaluation is limited to tree-based models that only do binary classification. How would the attack perform on more complex models, such as deep neural networks, or on multi-class classification tasks? Would the attack still be effective in these scenarios?
- **Impact of Training Dataset Size**: How does the size and representativeness of the training dataset affect the success of the attack? How many training data points are needed for the adversary to successfully launch this attack?

---

> ### Author Response · Authors · 2025-11-27
> **Response to reviewer DHwq (1)**
>
> We thank the reviewer for the insightful questions.
>
>
> > 1. **Threat Model Clarification**:  The threat model could be better defined. Specifically, what are the capabilities and limitations of the adversary? Why such capabilities are realistic?
>
> We are happy to clarify our threat model. At a high level, the goal of cryptographic model certification is to verify that a certain property (e.g., fairness, accuracy), holds for a given model. The adversary is the model provider, while the defender is the external auditor (who is not allowed to learn private information about the model/training data, e.g., model parameters).
>
> ## Capabilities of the adversary
> The adversarial model provider aims to pass the audit even if its model does not satisfy the audited property. In cryptographic protocols, adversaries may arbitrarily deviate from the protocol specification. In the context of model certification, this means the adversarial provider may:
>  1. Choose any training data (denoted by $S_{train}$).
>  2. Arbitrarily set the model parameters (determining $h$).
>
> Additionally, at some point the adversary receives an audit dataset $S_{audit}$ and then engages in a zero-knowledge proof protocol with an auditor.
>
> The crux of our threat model is that it formalizes the adversary's objective (e.g., making an inaccurate model pass an accuracy audit or an unfair model pass a fairness audit) in the context of model certification, while explicitly specifying whether the adversary learns $S_{audit}$ **before** or **after** performing actions 1 and 2 above.
>
> Now, let us consider the potential countermeasure proposed by the reviewer:
>
> > If the training dataset is pre-determined and fixed, then by hashing the training data, the adversary cannot freely choose the training data to manipulate the model.
>
> The viability of this alternative solution depends on the scenario, which we elaborate on below. **We then conclude that our proposed protocol template in Section 5 serves as the most efficient and provably secure solution.**
>
> ## Scenario 1: Adversary learns $S_{audit}$ *before fixing the training data $S_{train}$ and model $h$*
> Numerous prior works in the field (see Table 1) assume that the adversary is capable of knowing the audit dataset (we recently consulted with a large US firm conducting (non-cryptographic)  auditing of credit decision models and confirmed that this is standard practice for them as well).
>
> Hashing does not help those prior works that assume knowledge of the audit dataset, as the adversary can simply craft the training data based on the audit dataset *before* producing the hash. Thus, our attack from Section 4 is still applicable.
>
> ## Scenario 2: Adversary learns $S_{audit}$ *after fixing the training data $S_{train}$*
> If the adversary is required to fix the training data $S_{train}$ in advance by outputting a hash of it, one could prevent our attack by requiring the provider and auditor to perform both a _zero-knowledge proof of training_ (confirming that a model was honestly trained on the given data) and a _zero-knowledge proof of the audited property_ (e.g., fairness or accuracy). In this setup, the adversarial provider cannot make the model depend on $S_{audit}$ because it must adhere to the model derived from the pre-determined $S_{train}$, thereby preventing the training-data forgery described in Alg. 1.
>
> However, cryptographic proofs of training are notoriously expensive compared to proofs of the audited property (e.g. [a]). This motivates us to avoid proofs of training and instead adopt the simpler solution in Section 5.
>
>
> [a] Sun et al. Zkdl: Efficient zero-knowledge proofs of deep learning training. 2024.
>
> ## Our solution: Adversary learns $S_{audit}$ *after fixing the model $h$*
> As detailed in Section 5, our secure protocol template requires that the audit dataset is sampled from the distribution and revealed to the model provider after the model $h$ is fixed.
>
> Unlike the alternative above, _our approach does not require an expensive proof of training or a hash of the training data._ Our crucial observation is that this template can be made provably secure using only a zero-knowledge proof of the audited property, which is comparatively lightweight. Intuitively, the template remains sound even without forcing the attacker to follow a legitimate training process, because $S_{audit}$ is sampled independently of $h$, which is sufficient to check the true accuracy, fairness, and related properties of any $h$.

---

> > ### Author Response · Authors · 2025-11-27
> > **Response to reviewer DHwq (2)**
> >
> > > 2. **Generalizability of the Attack**: The evaluation is limited to tree-based models that only do binary classification. How would the attack perform on more complex models, such as deep neural networks, or on multi-class classification tasks? Would the attack still be effective in these scenarios?
> >
> > The attack in Algorithm 1 (and similar attacks referenced in section 7) generalizes trivially to multi-class classification. We have also conducted experiments on deep neural networks with a modified attack framework: over 10 runs, we achieved an average of 70% accuracy on test data and 89% accuracy on audit data. While there is less accuracy separation than our decision tree results, we assert that it provides a proof of concept, showing that a malicious service provider can manipulate data to perform audit evasion even on other model types.    We stress, however, that our attack framework was designed with decision trees in mind in an effort to provide theoretical guarantees, and that tailor-made attacks against neural networks are likely to find more success.
> >
> >
> > > 3. **Impact of Training Dataset Size**: How does the size and representativeness of the training dataset affect the success of the attack? How many training data points are needed for the adversary to successfully launch this attack?
> >
> > The attack in Algorithm 1 uses a training set with $|S_{audit}| \cdot d$ points. When attempting to learn more nuance behavior than rejecting every point, however, it is helpful (but not necessary) to add in additional training data. In Figures 2 & 3, we use roughly 70% of the data included in the dataset (with the rest held out for audit data and test data); we also scale how much audit data we use with an attack parameter $\alpha \in [0,1]$ to have a training set of variable size: $\alpha \cdot |S_{audit}| \cdot d+k$ where $k$ is the amount of extra training data included. Various settings of $\alpha$ are displayed across the $x$-axis of these figures, and the attack effectiveness is displayed on the $y$-axis.

---

### Official Review · Reviewer_hdTj · 2025-10-31

**Soundness:** 2
**Presentation:** 3
**Contribution:** 2
**Rating:** 4
**Confidence:** 3

**Summary:**

The paper studies data-dependent vulnerabilities in cryptographic auditing methods for machine learning models. The paper proposes an attack strategy that passes cryptographic certification while undermining the goals of those certifications for real-world performance. The paper also conducts experiments to evaluate the performance of the proposed method.

**Strengths:**

[+] The paper shows that that a model provider can attack many cryptographic model certification schemes by forging training data.

[+] The paper formalizes the guarantees an auditing framework should achieve.

[+] The paper introduces a generic protocol template that meets these requirements.

**Weaknesses:**

[-] The problem scope studied in the paper seems limited. The paper establishes the attacks for decision trees, which restricts the generalizability of the proposed method in the paper. It remains unclear whether the proposed attack framework can be extended to other models, such as different variants of decision trees, ensemble methods (e.g., random forests), deep neural networks, and diffusion models.

[-] The investigation on the potential detection defenses is insufficient. The paper considers the detection methods that check whether the training data and audit data were from the same distribution. On the other hand, the paper claims that the proposed attack can achieve only 30% accuracy on new samples from the same distribution. However, for the attack proposed in the paper, it can be easily detected by using accuracy on new samples from the same distribution, where a large performance drop would serve as an indicator of compromise. Additionally, the paper overlooks a discussion of existing detection defenses [1,2,3] against data poisoning attacks, many of which do not rely on distributional similarity, and propose different detection methods.

[-] The discussions of data poisoning attacks are not sufficient. The paper claims that data poisoning attacks involves subtle, often small-scale perturbations to the training data. However, there are many other forms of poisoning attacks beyond such perturbations, including backdoor poisoning that can compromise models with visible data features (e.g., stickers). Moreover, it remains unclear whether the paper considers label perturbations or only feature-level manipulations. The authors should also clarify how the proposed method performs under large-scale perturbations, discuss its strengths and limitations in such scenarios, and specify any constraints on the poisoning ratio within the training data.

[-] The paper fails to discuss existing during-training defenses and post-training defenses against traditional data poisoning attacks. Furthermore, the paper does not analyze whether these existing defense strategies could mitigate the proposed attack, which is important for understanding the novelty and real-world robustness of the attack.

[-] The paper does not provide an analysis of the computational complexity of the proposed method. Additionally, the distribution similarity–based detection lacks any discussion of its computational cost or scalability when applied to large datasets or high-dimensional features. Moreover, it would be beneficial to compare the computational efficiency of the proposed attack with traditional data poisoning attacks.

Reference.

[1] De-Pois: An Attack-Agnostic Defense against Data Poisoning Attacks, 2021.

[2] Deep Probabilistic Models to Detect Data Poisoning Attacks, 2019.

[3] A survey on data poisoning attacks and defenses, 2022.

**Questions:**

[1] Could the authors discuss how to perform sampling for $S_{audit}$? What criteria or strategies are used to determine which samples are selected, and how is the sample size decided? Additionally, it would be helpful to discuss whether there is any potential overlap between the audit samples and the training data, and if so, how such overlap might affect the validity of the detection process.

[2] Could the authors clarify which steps in the proposed method can be omitted and under what conditions? For the omitted steps, it would be better to discuss the resulting characteristics of $ f $ and how such omissions may influence its performance or robustness. Can the proposed method still work in Shamsabadi et al. (2022)?

[3] Are there any different impacts of different thresholds (e.g., 0.95) on the proposed method? How to optimally define these thresholds, and what is their sensitivity to variations? How would the parameter $\varepsilon$ influence the effectiveness and stability of the proposed method?

[4] Could the authors explain why it is unlikely that other statistical tests will be effective in detecting the attack?

---

> ### Author Response · Authors · 2025-11-27
> **Response to reviewer hdTj (1)**
>
> We thank the reviewer for raising these points.
>
> # Response to weaknesses
> > W1. Generalizability of the proposed method to other model types.
> >
> Decision trees are a commonly used, theoretically well-understood machine learning method. By focusing on decision trees we are able to provide concrete theoretical results such as Theorem 1, which guarantees isolation of points in $S_{audit}$ on the decision surface. For less theoretically understood models such as neural networks, heuristic approaches such as classical poisoning attacks may achieve better results. We reserve adaptation of these attacks to our setting as an interesting direction for future work. For completeness, we conducted additional experiments applying our existing attack method to neural networks. Over 10 runs, we achieved an average of 70% accuracy on test data and 89% accuracy on audit data. While there is less accuracy separation than our decision tree results, we assert that it provides a proof of concept, showing that a malicious service provider can manipulate data to perform audit evasion even on other model types.
>
>
> > W2. The attack proposed in the paper can be detected by using accuracy on new samples.
> >
> The reviewer is correct that the attack in Algorithm 1 can be detected by using new samples - indeed, this is part of our solution (see Section 5). However, to satisfy the requirements of the cryptographic model certification setting, such detection must be done securely to ensure that a) the corrupt model provider does not switch or alter the model during the audit, and b) without leaking any information about the model apart from the fact that the model satisfies the audited property.
>
> Crucially, as we discovered in our case studies (see Table 1, Section 6, and Appendix D), numerous recent works published across top venues in machine learning and security do *not* explicitly consider verification on new data. Thus these certification methods are vulnerable to the straightforward attack described in Algorithm 1. To the best of our knowledge, our work is the first to formalize cryptographically secure audits with respect to an audit data *distribution* rather than a single audit dataset. Thus our work strengthens the cryptographic model certification threat model against data manipulation, enabling provably secure audits that incorporate evaluation on fresh samples without compromising security or correctness guarantees.
>
> Please let us know if there are further questions on this point.

---

> ### Author Response · Authors · 2025-11-27
> **Response to reviewer hdTj (2)**
>
> > W2, W3, W4. On data poisoning attacks and differences to our setting.
> >
> Our work is focused on **cryptographic model certification**, which is in a different setting and has a different threat model from data poisoning. In our scenario, the
>     model provider is the *adversary*, while in traditional data poisoning the model provider is the *defender*. This results in an entirely different set of adversarial capabilities, and imposes additional constraints on the defender. In particular:
>
>  - **The adversary in our setting can perform arbitrary manipulations to the model and the training data.** This in particular includes both label perturbations and feature-level manipulations. It further encompasses adversaries corrupting **all** training data, i.e., there are **no** constraints on the poisoning ratio. Finally, the adversary may even corrupt the model parameters **directly**. Note that simply asking a model provider to execute a certain protocol (including a data poisoning defense mechanism) is **not sufficient** -- a malicious model provider could choose to simply ignore this instruction. Our goal is to design an audit such that—even under these powerful adversarial capabilities—if the model passes the audit, then it must genuinely satisfy the audited property (e.g., a fairness or accuracy threshold).
>
>  - **Both the model and the training data are hidden from the defender.** As in traditional data poisoning the defending party is the model provider, they are allowed to view both the model and the training data. In cryptographic model certification, the defending party is an *external auditor*, who is not allowed to view the (potentially propriatery/private) model/training data. This imposes additional constraints on the defender, who is tasked with assessing the validity of the model *without ever seeing it*, and is the reason why known solutions with provably secure guarantees (including ours) crucially rely on cryptographic techniques.
>
> To summarize, as our adversary is strictly stronger than in traditional poisoning—able to arbitrarily modify all data and even directly set model parameters—our framework naturally covers all forms of classical poisoning, including large-scale perturbations, label corruption, and visible backdoor triggers. However, our goal is fundamentally different: we do not attempt to detect or mitigate poisoning during training, but instead ensure that any model that passes the audit must satisfy the audited property, regardless of how it was produced. We will add an expanded discussion about differences to data poisoning in the final version of the paper.
>
>
> > W5. Computational complexity of the proposed method.
>
> We will clarify the computational complexity of our proposed method by adding the following text to the methods section:
>
> > > The computational complexity of our attack is linear in the size of the audit dataset, i.e. $\mathcal{O}(|S_{audit}|\cdot d)$. This is clear since the training set $S_{train}'$ contains two additional points in each dimension per point in $S_{audit}$.
>
> This means that our adversary runs in polynomial time, a standard assumption for many cryptographic adversaries. Further, limiting the computational complexity of the attacker could result in a security model that can defend against adversaries with less compute, but breaks against more computationally powerful ones. Thus, we do not consider computational efficiency to be a germaine axis of comparison for our work, given that our attack runs in sub-exponential time.
>
> # Response to questions
> > Q1. How to perform sampling for $S_{audit}$?
>
>  - **If the reviewer is asking about the secure protocol template in Section 5**: Following standard practice, $S_{audit}$ should be sampled i.i.d. from the distribution in question. Appendix E.2.1 and E.2.2 provide formal guidelines for setting the sample size to realize a sound auditing protocol with negligible adversarial advantage. For instance, to provably guarantee the attacker's cheating probability $<2^{-\lambda}$ in accuracy auditing, it would be sufficient to set $n\delta^2=\lambda$, where $n$ is the sample size and $\delta$ is the gap between true accuracy and sampled accuracy.
>  - **If the reviewer is asking about the attack game in Section 3**: Our data forging attack makes no assumption on the process in which $S_{audit}$ is sampled. In fact, our Theorem 1 formally guarantees that the attack against decision trees is successful regardless of input $S_{audit}$.

---

> > ### Author Response · Authors · 2025-11-27
> > **Response to reviewer hdTj (3)**
> >
> > > Q1. Does an overlap between the audit samples and the training data affect the validity of the detection process?
> >
> >  - **If the reviewer is asking about the secure protocol template in Section 5**:  While $S_{audit}$ and $S_{train}$ might have a potential overlap, our analysis of a secure protocol template (see Section 5 & Appendix E) implies that the auditing scheme provably satisfies our soundness notion as long as $S_{audit}$ is freshly sampled i.i.d. after the model $h$ is fixed.
> >
> >  - **If the reviewer is asking about the attack game in Section 3**: Our data forging attack works by incorporating $S_{audit}$ into $S_{train}$. As we discuss in L.313 and Appendix C, such an overlap can be exploited to evade additional detection measures such as $t$-test, as the attacker can cause $S_{audit}$ and $S_{train}$ to appear as if they ware drawn from the same distribution.
> >
> >
> >
> > > Q2. Which steps in the proposed method can be omitted and under what conditions? Can the proposed method still work in Shamsabadi et al. (2022)?
> >
> >  We respond assuming that the reviewer is referring to the sentence starting with "Depending on $f$" on L.199-202. (If we have misunderstood the question, we would appreciate it if the reviewer could clarify.)
> >
> > The paragraph in question is reviewing previous methods, not our proposed framework.
> > We simply meant that the syntax we give in that section is kept intentionally general to capture a wide variety of prior works. As a result, some existing protocols do not include every step of our unifying syntax.
> >
> > For example, Zhang et al. (2020) verify the accuracy of the model $h$, and their predicate $f$ only needs to check that $h$’s empirical accuracy on $S_{\text{audit}}$ exceeds a given threshold. Hence, their protocol does not require a commitment to the training data (since it is never used).
> >
> > We emphasize that we **do not** suggest omitting any **protocol** steps -- neither in prior protocols, nor in our own.
> >
> >
> > With regards to Shamsabadi et al. 2022, their certification method allows the service provider to choose the audit dataset, and thus our fairness attack (Figure 3) straightforwardly applies -- we discuss this in Appendix D3.
> >
> >
> > > Q3. How do different thresholds (e.g., 0.95) impact the proposed method?
> >
> > As illustrated in Figures 2 and 3, the attacker can pass auditing with nearly any threshold by tuning the "attack parameter", which controls the proportion of audit data points included in the training data and the proportion of the initial training data labeled maliciously. Thus the attack is viable regardless of the threshold set by the auditor. For example, to pass an accuracy audit with a threshold of $0.75$, it is sufficient to set the attack parameter to $\approx{}0.6$.
> >
> > > Q3. How would the parameter $\epsilon$ influence the effectiveness and stability of the proposed method?
> >
> > $\epsilon$ controls the size of the isolated 'chunks' of the decision surface that contain each audit point. Thus in principle increasing $\epsilon$ could reduce the effectiveness of the attack on some distributions, while decreasing $\epsilon$ could improve it. However, we conduct additional experiments which demonstrate that in practice the attack is highly stable to varied settings of $\epsilon$ for the ACSEmployment dataset:
> >
> > ### $\epsilon=2$
> > | Attack Parameter | Audit Acc. | Test Acc. |
> > | -------- | -------- | -------- |
> > | 0        | 0.7527   | 0.7568   |
> > | 0.1      | 0.7349   | 0.7013   |
> > | 0.2      | 0.7091   | 0.6504   |
> > | 0.3      | 0.7253   | 0.5995   |
> > | 0.4      | 0.7314   | 0.5455   |
> > | 0.5      | 0.7506   | 0.4941   |
> > | 0.6      | 0.7732   | 0.4385   |
> > | 0.7      | 0.8120   | 0.3876   |
> > | 0.8      | 0.8638   | 0.3446   |
> > | 0.9      | 0.9284   | 0.3063   |
> > | 1        | 0.9967   | 0.2670   |
> >
> >
> > ### $\epsilon=3$
> > | Attack Parameter | Audit Acc. | Test Acc. |
> > | -------- | -------- | -------- |
> > | 0        | 0.7557   | 0.7588   |
> > | 0.1      | 0.7364   | 0.7055   |
> > | 0.2      | 0.7231   | 0.6547   |
> > | 0.3      | 0.7202   | 0.6045   |
> > | 0.4      | 0.7329   | 0.5528   |
> > | 0.5      | 0.7463   | 0.4986   |
> > | 0.6      | 0.7767   | 0.4460   |
> > | 0.7      | 0.8160   | 0.3943   |
> > | 0.8      | 0.8669   | 0.3514   |
> > | 0.9      | 0.9277   | 0.3082   |
> > | 1        | 0.9954   | 0.2709   |
> >
> >
> >
> > > [4] Could the authors explain why it is unlikely that other statistical tests will be effective in detecting the attack?
> >
> > Theorem 2 proves that a broad family of statistical measures cannot be used to detect our attack. Lemma 2 and corollary 1 specifically tailor this result to the case of the t-test. It seems likely that many other statistical tests follow the same pattern, as increasing the number of copies of the audit data in the training data shrinks the statistical distance between the training data and the audit data, though we reserve this exploration for future work.

---

### Official Review · Reviewer_NhAG · 2025-10-31

**Soundness:** 2
**Presentation:** 2
**Contribution:** 1
**Rating:** 2
**Confidence:** 4

**Summary:**

The paper proposes an attack framework designed to bypass model auditing systems by exploiting knowledge of the audit dataset. Specifically, it assumes that if a model provider knows the audit dataset in advance, they can optimize the model to perform extremely well on this dataset while performing poorly on unseen data. The attack is termed as data forging, where the training data or model behavior is manipulated to pass the audit, thus undermining the integrity of the auditing process.

The authors formalize this setup as a game between the auditor and the model provider. Experiments are conducted on classification and text generation benchmarks, showing that the forged models achieve high audit accuracy but low generalization performance. The paper claims this demonstrates a critical weakness in current audit mechanisms.

**Strengths:**

1. The paper focuses on model auditing, an increasingly important area for responsible AI deployment and accountability.

**Weaknesses:**

1. The paper assumes that the attacker knows the audit dataset in advance, making the threat model fundamentally trivial. If the attacker has access to the audit data, they can directly fine-tune or overfit the model to achieve near-perfect audit performance, making the auditing process meaningless. This scenario does not reflect real-world auditing practices, where audit datasets are typically held confidentially or sampled independently from held-out data.

2. The proposed commit–sample–prove protocol assumes that the audit dataset is sampled after model commitment, directly contradicting the threat model where the attacker already knows the audit data. This makes the proposed solution largely circular—it fixes the attack only by removing the condition that enables it. Taken together with point 1, the overall contribution feels conceptually weak: the paper defines an unrealistic vulnerability and then proposes a remedy that depends on changing the assumption itself. In practice, well-designed audit processes already ensure that audit datasets remain secret or are generated after commitment, so the identified issue is unlikely to arise.

3. The paper assumes that the service provider (adversary) has full knowledge of the audit dataset and constructs an alternative training dataset ( $S'_{\text{train}}$ ). However, the construction of such adversarial data is extremely limited—it simply minimizes the loss on the audit dataset and its perturbed variants, where the perturbations are created by adding small random noise to the features (algorithm 1). In essence, the adversary’s objective is merely to enforce certain properties on the given audit dataset, which is conceptually similar to standard data augmentation or fairness optimization techniques. As such, the paper lacks substantive algorithmic novelty and does not introduce any fundamentally new mechanism or optimization idea.

4. Similar to point 3, the threat model used in the data-forging attack is comparable to, or even weaker than, that of classical data poisoning attacks, where the adversary injects a small amount of malicious or targeted data into the training process to influence model behavior on specific subsets of the distribution [1]. The proposed data-forging scenario simply assumes access to the audit dataset and optimizes directly on it, which is a less realistic and less technically challenging setting than existing poisoning-based attacks. Consequently, I do not view data forging as introducing a fundamentally new risk beyond what prior poisoning literature has already established.

[1] Jagielski, M., Oprea, A., Biggio, B., Liu, C., & Chen, B. (2021). Subpopulation Data Poisoning Attacks. IEEE Symposium on Security and Privacy (S&P), 2021.

**Questions:**

See the weaknesses. My key concerns are:

1. Please justify the assumption that the model provider knows the audit dataset in advance. How realistic is this in any practical auditing framework? If the attacker already knows the audit data, why is a new attack framework needed? Isn’t this just overfitting to a known test set?

2. Can you provide an alternative attack setting where the provider only has partial or approximate access to the audit dataset (e.g., distributional hints)? Evaluating partial-knowledge attacks would make the scenario more realistic and non-trivial.

3. How does this work differ from poisoning attacks technically? Please contrast your threat model, attacker capabilities, and method with representative poisoning papers (e.g., Jagielski et al., 2021) and clarify what, if anything, is novel in your adversary’s optimization or data-construction method.

---

> ### Author Response · Authors · 2025-11-24
>
> We thank the reviewer for the insightful questions. They clarify important differences between our threat model and previous work.
>
> >Q3. Contrast your threat model with poisoning
>
> Our threat model is specific to attacks on **cryptographic model certification**, an emerging suite of methods that verify model properties to an auditor while keeping parameters and training data confidential. The key difference from classical poisoning (e.g. Jagielski et al.) is that **the adversary is the AI/ML service provider** rather than an external party that injects corrupted data. This means that the adversary gets *unilateral control* over the data and model prior to certification. Adversaries in our setting can thus mount **stronger** attacks, while confidentiality creates **additional constraints** for defenders.
>
> We will revise the Related Work to emphasize two major points:
>
>   - **Unbounded Proportion of Corrupted Data.** In classical poisoning we assume there is some honestly generated data in the training set, but our adversaries may corrupt the *entire dataset*. Our setting is thus more general: poisoning adversaries are a strict subset of model certification adversaries.
>  - **Adversary Can Disable Classical Defenses (by choosing not to execute them).** Since the adversary has full control of the data and model prior to certification, they can evade classical poisoning defenses by simply choosing not to execute them. Meanwhile confidentiality makes it difficult for an external auditor to verify whether a poisoning defense has been deployed.
>
> > Q3, Q1. Clarify what is novel in your adversary’s optimization or data-construction method. Why is a new attack framework needed? Isn’t this just overfitting?
>
> Table 1 & App D show that many emerging works in cryptographic model certification appear unaware of how data manipulation can induce vulnerabilities. We provide Alg 1 *not* to claim that overfitting is a new idea, but rather to display a concrete attack on this existing paradigm. To highlight that this is just one of many vulnerabilities under our threat model, we expand on our case study of Shamsabadi et al. 2024 in App D9 with a concrete attack:
>
> **Data forging attack on certificates of Differential Privacy**
> **Inputs**: dataset $S$, set of targets $T \subset S$, duplicity $n$.
> **Output**: forged dataset $S'$
> 1. $S' \gets S$
> 2. for each target $x \in T$:
>     - repeat $n-1$ times:
>         - uniformly sample small $\epsilon \in \mathcal{X}$
>         - add $x+\epsilon$ to $S'$
> 3. return $S'$
>
> This data forging attack enables an adversary to certify a model as differentially private despite severely degrading its privacy guarantees. By adding many copies of points close to each target $x$, the adversary creates a dataset that is inadequately protected by the amount of noise added in the 'certified' differentially private training.
>
> > W1, Q1. Justify the assumption that the model provider knows the audit dataset in advance.
>
> > W2. In practice, well-designed audit processes already ensure that audit datasets remain secret or are generated after commitment.
>
> Many previous works in model certification use **public** audit datasets (App D). We also surveyed real-world practices by consulting with a large US firm conducting (non-cryptographic) black-box auditing of credit decision models. They confirmed that _the auditing dataset is typically picked by the model providers, with no mechanism like commit-sample-prove to ensure the dataset is determined after training_. This is because companies rely on mutual trust in practice, and threat models involving misbehaving providers are often not considered.
>
> Given these vulnerabilities, we believe it is important to provide formal guidance to practitioners by precisely specifying the order in which commitments to the model and audit data are generated. However, to the best of our knowledge no previous work has proven a statement similar to Thm 3 & App E.2, which rigorously proves that a commit-sample-prove protocol satisfies a suitable soundness notion w.r.t. real-world distribution using only the ZK proof of accuracy/fairness. We note that achieving this result required carefully combining security properties of commitments, ZK proofs, and statistical learning theory.
>
> > Q2. partial or approximate access to the audit dataset [...]
>
> Figures 2 & 3 partially address this question. The attack parameter consists of two components: (i) how much training data is label-flipped, and (ii) how much audit data is known to the service provider at training time. We run additional experiments that isolate the second component:
>
> | % Audit Data Known | Audit Acc | Test Acc |
> | ------------------ | --------- | -------- |
> | 0                  | 0.23      | 0.23     |
> | 20                 | 0.39      | 0.24     |
> | 40                 | 0.55      | 0.25     |
> | 60                 | 0.69      | 0.25     |
> | 80                 | 0.84      | 0.26     |
> | 100                | 0.99      | 0.26     |

---

### Author Response · Authors · 2025-12-03
**Discussion & Revision Plan**

We once again thank the reviewers for their constructive comments on our work. Incorporating their feedback will improve our manuscript. Here we will discuss key strengths by the reviews, summarize the additional results obtained in the discussion period, and formulate a revision plan.

## Strengths
The reviewers have noted the following strengths of our submission:

- **Relevance and Significance of the Topic**: The paper tackles ML model auditing, whose safety and security analysis would be significant for responsible AI deployment and accountability.

- **Data Forging Attack**: The paper introduces novel attack vectors against existing cryptographic model certification schemes that remain undetected by auditors, highlighting the severity of data-dependant vulnerabilities.

- **Formalization**: The paper studies and formalizes the guarantees a secure auditing framework should achieve.

- **Secure Protocol Template**: The paper introduces a generic protocol template that meets the formalized requirements.

As our original submission already stated our key contributions across the above dimensions, we believe that the reviewers agreed on the significance of our work.


## Additional Results and Experiments
During the discussion period, we conducted the following additional experiments that further substantiate the generality and strength of our proposed attack. We summarize these results and elaborate on how we plan to incorporate them into the revision:

- Reviewers hdTj and DHwq both asked whether our attack generalizes to other model types. In additional experiments, we showed that **the attack in Algorithm 1 generalizes to neural networks**, achieving an average of 70% accuracy on test data and 89% accuracy on audit data. We will add this, and experiments on additional benchmark datasets, to the results section. We will also explicitly note that adapting classical backdooring attacks [a] to the data forging setting is a promising future direction for improved attacks on neural networks.
- Reviewer NhAG suggested that considering attackers with only **partial knowledge of the audit set** would broaden the scope of our work. Figures 2 & 3 partially capture this notion, and in our response we presented experiments that vary the attacker's knowledge of the audit set as an isolated variable. The results confirm that the attack has **significant effects even if the audit set is only partially known.**
- With additional experiments, we addressed Reviewer hdTj's questions about how the $\epsilon$ parameter in Algorithm 1 would affect stability of the attack. Our results showed that **the attack achieves high performance under a variety of $\epsilon$ settings.** We will add these experiments to the Appendix.

[a] T.D. Nguyen et al. Backdoor attacks and defenses in federated learning: Survey, challenges and future research directions. 2024.

## Revision plan
We thank the reviewers for acknowledging the importance of our work, and for their highly constructive feedback. Our paper formalizes a threat model in the interdisciplinary space between cryptography and machine learning. We posit that the bulk of the questions raised in the reviews stem primarily from difficulties of communication across this interdisciplinary boundary, rather than issues with the merit of the work itself. As such, clarifying the Reviewers' questions will help make our contributions more clear to all readers.

We will revise the manuscript to fully address the reviewers' questions, including (1) additional experiments stated above, (2) threat model clarification, and (3) applicability and practical relevance of the attack, and other miscellaneous improvements.

---

### Meta-Review · Area_Chair_hkeX · 2026-01-02

**Summary:**

All in all, the authors demonstrate a valid problem with the current black-box auditing techniques of ML systems. They motivate it from a cryptographic and machine learning point of view, and propose a reasonable cryptographic defense for the issue. The main issue with the paper is that it targets heavily a cryptographic audience, which leads to 2 major negative consequences. One is that the reading is made really hard for the ML audience, as while ML conclusions are given very detailed (or even too detailed) high-level explanations, the cryptographic conclusions lack the same treatment. The second is that much of the well-known basic ML knowledge is rediscovered in the cryptographic setting for no apparent reason. This leads to theoretical analysis that is unnecessarily specific to individual classifiers or statistical tests, as well as redefinitions of many common and well-researched concepts in theoretical ML. This significantly lowers the value of the proposed attack and defense method for the ML community. While I **do not** suggest the paper cannot be written to make it much more useful to the ML community, I do think that the current version of the paper is **not suitable** for an ML conference. If authors demand to stay closer to the current framing of the work, I would suggest they resubmit to S&P or CCS. That said, I would suggest that a much better course of future action will be to adopt more ML machinery and terminology in future versions of the paper, making it suitable and useful for both communities.

**Reviewer Concerns:**

**Outstanding Reviewers' Concerns**
- **Pointing out an obvious attack vector (Reviewer NhAG Weakness 1)**
I agree both with the reviewer **AND** the authors on this point. On the one hand, the authors are right. Many current systems seem to be unaware of the attack vector pointed out in the paper, especially those coming from the Security/Cryptography community. This suggests that visibility on the issue is important and that the research community is the one that can drive the effort by systematizing the issue. On the other hand, I think the reviewer is right too - the problem pointed out by the authors is ultimately an **ML issue** that is well-studied in the ML community. In particular, this is the problem of generalization and overfitting. Here, a vast body of existing studies on generalization, overfitting, probably approximately correct (PAC) learning, and function sensitivity makes the point obvious to ML researchers and, thus, of less use to the ML community. Somehow, it seems to me that this study, thus, has more use in a cryptography venue such as S&P or CCS. That said, I think the problem is made vastly worse by the fact that the authors (who, based on the writing, I assume are coming from the Cryptography community) have not adopted the ML language in their paper. Examples of this:
    1. F˜-relaxed Knowledge Soundness is essentially the same definition as PAC Learning, but in the context of ZKPs
    2. Algorithm 1 just says that if you branch decision trees enough to fit the training data perfectly, you achieve overfitting. That is crystal clear to any ML researcher. This is caused by the sensitivity of the family of functions represented by decision trees with unbounded branching. Something PAC learning controls by limiting the expressiveness of the function family by limiting branching. Moreover, that same point can be made for large families of classifiers by just borrowing existing proofs on overfitting from the ML community. Please also look into VC-dimension, as it might be a way to do the extension in a classifier-independent way.
    3. Sampling in Audit in Section 5 is underspecified. It seems to me that a proper ML/statistics argument will need to specify that S_audit needs to be independently uniformly sampled from D, and that if other sampling schemes are used, the protocol will not provide the required level of protection. Again, an ML viewpoint on this will not allow such an important omission.
    4. I even think that some of the open questions the authors pose regarding statistical recognition with respect to the training set have been partially addressed in the ML community on backdoor detection, or at least people have developed some tools to address them. For example, check [1].
I think the paper will therefore benefit from using more ML language w.r.t. PAC learning, overfitting, etc, as it will also allow the authors to broaden its scale significantly, make it more general, and reuse 40 years of existing results in the field.
- **Algorithm 1 is not novel (Reviewer NhAG Weakness 3)**
I agree with the Reviewer, the method seems to be one algorithm (of, I think, many available) that makes decision trees with unbounded depth to overfit.
- **Sampling of S_audit in Section 5 (Reviewer hdTj Question 1)**
As I already explained above, it seems to me that a proper ML/statistics argument will need to specify that S_audit needs to be independently uniformly sampled from D, and that if other sampling schemes are used, the protocol will not provide the required level of protection.
- **Threat Model Clarification (Reviewer DHwq Question 1)**
I actually think this remains very much underspecified in the current paper version. I will argue that the authors should outline exactly the capabilities of the attacker and the defender and possibly even distinguish different families of attackers based on knowledge and access - e.g. Partial Knowledge vs Full Knowledge of S_audit and Full access (possibly changing the weights completely) vs Training dataset modification (which might be detectable by a combination of proof of training + statistical tests).

**Outstanding AC Concerns**
- **Focus on Cryptography Language that makes it hard to read**
While I am versed in both languages (despite coming from the ML side), I must say that the paper currently strongly favours Cryptography readers compared to ML readers in terms of the high-level explanations of the techniques used. Simple concepts and the author's results from the ML side seem to be quite decently explained at a high level. In contrast, the Cryptography writing makes no attempts at high-level justifications and explanations, burning a possible bridge to the understandability from ML community members.
- **More general defense**
I felt that the presentation of the (Commit, Prove, Audit)-protocol is somewhat constraining compared to the set of possible implementations of the idea. For example, it is not clear to me what properties are required by the sampling of S_audit (e.g., are other sampling algorithms outside of iid permitted), can multiple rounds of challenges through multiple sampled S_audit datasets be beneficial? I think making the protocol more generic will allow further instantiations that can further improve sample efficiency or runtime, which is practically important for the adoption of the algorithm to realistic ML systems.
- **VC-dimension**
Are there properties of the VC-dimension of the classifier that will allow us to determine how vulnerable it is to the proposed attack vector?
- **The authors say that in their limited experiments NNs are less vulnerable**
In practice, NNs are usually **more** susceptible to overfitting compared to trees; thus, I think that they will likely end up more vulnerable to the proposed attack vector if the attack algorithm is well designed. To this end, I feel it is important to show in the next revision of the paper.
- **Impact of the auditing distribution**
I think it will be useful for the authors to comment on how properties of the auditing distribution D affect the attack vector and their proposed defense protocol, as this can, for example, inform an auditor how to better choose the S_audit dataset size.
- **More background/motivation text is needed about auditing**
The authors do not spend enough space in the paper motivating the i) need for auditing, ii) the need for black-box auditing with privacy guarantees, and iii) the complexity of auditing leading to reliance on public datasets. On the last point, I think [2] might help.
[1] https://proceedings.mlr.press/v139/hayase21a/hayase21a.pdf
[2] https://arxiv.org/pdf/2410.07959

**Reviewer Scores:**

Reviewer NhAG - I think the reviewer would likely not have been swayed by the provided rebuttal to change his grade, as the reviewer has a very strong ML view of the paper, and appears not to have understood the cryptographic and practical significance of the work. That said, I partially disagree with the reviewer on some of their points.

Reviewer hdTj - I think the reviewer would likely not have been swayed by the provided rebuttal to change his grade, as the reviewer really wants to better understand the connection to poisoning attacks, and I do not think the response of the authors is very satisfying (even though I understand the point that the two sets of attacker capabilities are different). That said the authors get credit for the additional experiments and clarifications they provide to some of the more technical questions of the reviewer.

Reviewer DHwq - I think this reviewer would have maintained the positive grade of the paper, given that the authors address all of the reviewer's questions well.

---

### Decision · Program_Chairs · 2026-01-26

Reject